# RINGLIGHT-GS: A COMPACT AND EXPRESSIVE FRAMEWORK FOR MODELING SCENE COLOR IN 3DGS

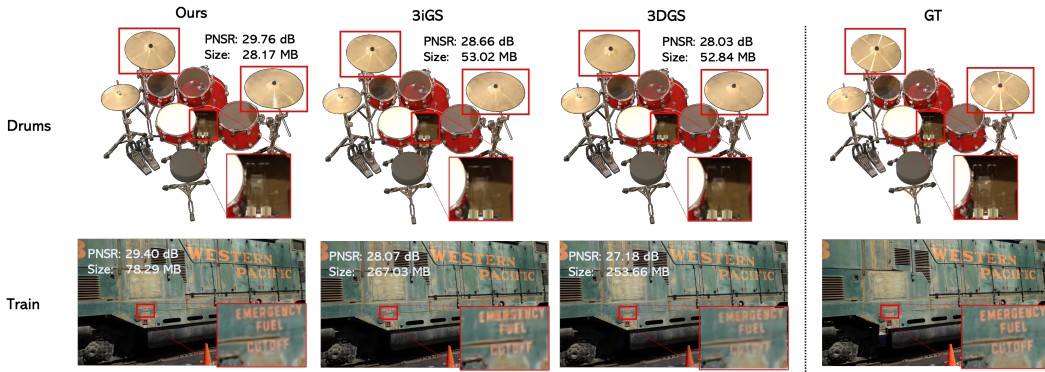

Figure 1: RingLight-GS enables compact and expressive modeling of high-frequency appearance.

## ABSTRACT

3D Gaussian Splatting (3DGS) achieves impressive novel view synthesis in real-time by directly rendering Gaussian primitives. However, it incurs substantial storage demands and struggles to model high-frequency, view-dependent appearance effects under complex illumination. We introduce RingLight-GS, a compact framework that effectively models scene color in 3DGS, delivering high-quality rendering under complex lighting while greatly reducing storage costs. The scene color is separated into a view-independent base color and a view-dependent residual color by disentangling static albedo from dynamic lighting, with the base color learning similarity to the 3DGS opacity. Specifically, the residual color is derived from view-dependent appearance features via a neural tensor ring regression model, influenced by spatial positions and viewing directions. Extensive experiments on synthetic and real-world datasets demonstrate that RingLight-GS consistently outperforms both NeRF-based and 3DGS-based baselines. It delivers sharper highlights, better material consistency, and lower perceptual error with minimal memory overhead.

## 1 INTRODUCTION

Novel view synthesis (NVS) generates new perspectives of a scene based on a collection of images and camera poses. It remains a fundamental challenge in 3D vision, propelling advancements in fields such as virtual and augmented reality (Deng et al., 2022; Rolff et al., 2023; Li et al., 2022), media generation (Zhou et al., 2018; Park et al., 2017), and autonomous driving (Pang et al., 2024; Ma et al., 2025; Zürn et al., 2024; Song et al., 2023). Neural radiance fields (NeRF) (Mildenhall et al., 2021) revolutionized the task of NVS by mapping a spatial location and a viewing direction to color and density using a multilayer perceptron (MLP). This powerful representation and its extensions have demonstrated impressive capability in generating photorealistic views by capturing complex 3D geometry and appearance (Barron et al., 2021; 2022; Fridovich-Keil et al., 2022; Müller et al., 2022). Despite their ability to significantly enhance rendering quality, their dependence on computationally intensive volumetric rendering limits their practicality in real-time applications.

3D Gaussian Splatting (3DGS) (Kerbl et al., 2023) renders scenes by rasterizing thousands of anisotropic Gaussians, with their positions, scales, opacities, and spherical-harmonic (SH) color coefficients learned jointly. This method avoids volumetric ray marching and allows for real-time NVS. However, 3DGS is constrained by its significant storage and memory requirements. In response, recent work (Tang et al., 2025; Tsai et al., 2025; Malarz et al., 2025) has proposed hybrid approaches that integrate neural representations into 3DGS, such as replacing explicit parameters with MLPs (Tang et al., 2025) or predicting view-dependent color and opacity (Malarz et al., 2025). Although these methods decrease storage requirements, they frequently compromise rendering quality due to the challenges in capturing high-frequency variations across spatially irregular Gaussians.

This highlights the need for more compact, expressive representations that balance Gaussians' storage efficiency with rendering quality. To improve the trade-off between compactness and fidelity, recent research (Chatziagapi et al., 2024; Tang & Cham, 2024) has investigated the incorporation of tensor decomposition into 3DGS, i.e., replacing thousands of independently stored SH color vectors with a shared, low-rank tensor basis. However, these methods still incur considerable storage overhead, and their ability to represent high-frequency view-dependent details remains limited. When revisiting 3DGS-related work, SHs are often deemed to encode complex lighting and color. The first degree represents diffuse color, and the higher degrees capture view-dependent colors. These account for 81% of the storage size required for a single Gaussian (Bagdasarian et al., 2025). When exploring the complex lighting and color in a standard 3DGS pipeline, a natural question arises: can we improve each 3D Gaussian's color fidelity to better capture details under complex conditions, thereby obtaining more compact 3DGS models and enhanced rendering quality?

Therefore, this paper introduces RingLight-GS, a novel framework that serves as an alternative to SH for modeling color in 3D Gaussians. RingLight-GS aims to factorize the appearance feature into two components: (i) a view-independent one that captures the intrinsic color of each Gaussian and (ii) a view-dependent residual color that encodes complex lighting effects, such as reflection and specular highlights. In particular, inspired by the success of tensor decompositions in preserving multi-linear structure (Saragadam et al., 2024; Luo et al., 2023) and implicit neural representations for compactly representing continuous inputs (Zhang et al., 2025b), we propose a neural tensor ring decomposition within the regression model to learn the high-frequency, view-related component of the illumination field.

As shown in Fig. 1, RingLight-GS produces sharper view synthesis results at a lower storage cost than previous baselines. The main contributions of this work are as follows:

- RingLight-GS builds on a standard 3DGS pipeline and replaces the per-Gaussian SH with an explicit base color and a shared Neural Tensor Ring (TR) illumination field over $(x, y, z, v)$, while keeping Gaussian geometry and rasterization unchanged.

- The proposed TR-based illumination representation provides a more favorable storage–quality–speed trade-off than SH-based 3DGS on standard NVS benchmarks, particularly under complex illumination with high-frequency view-dependent effects.

## 2 RELATED WORK

### 2.1 NOVEL VIEW SYNTHESIS

Novel view synthesis aims to render new scene views from posed images. NeRF (Mildenhall et al., 2021) employs an implicit neural representation to map 3D positions and view directions to volume density and color, enabling high-quality scene rendering. Mip-NeRF (Barron et al., 2021) uses cone-frustum integration instead of point sampling, reducing scale aliasing, parameters, and training time while maintaining volumetric rendering. Mip-NeRF 360 (Barron et al., 2022) introduces a space-contraction warp and distortion regularizer for unbounded 360° scenes, enhancing depth accuracy indoors and outdoors. Instant-NGP (Müller et al., 2022) replaces NeRF's sinusoidal encoding with a multi-resolution hash grid and a small fused MLP, enabling scene training in seconds for real-time playback. Tri-MipRF (Hu et al., 2023) decomposes the field into three orthogonal 2D mipmap planes with cones, achieving Instant-NGP's speed while retaining Mip-NeRF's anti-aliasing. Despite the major progress made, these solutions' reliance on volumetric rendering and dense ray sampling leads to high computational overhead, making them unsuitable for real-time applications.

Recently, 3DGS (Kerbl et al., 2023) uses anisotropic 3D Gaussians for explicit 3D scene representation and renders with a visibility-aware rasterizer, achieving real-time ($\geq$30 fps at 1080p) NVS and minute-level training. However, its pixel-center sampling still leaves aliasing and soft speculars. Analytic-Splatting (Liang et al., 2024) derives a closed-form pixel-area integral, such that each Gaussian footprint is anti-aliased across resolutions, removing jaggies without sacrificing 3DGS speed. Yet, its memory requirement remains comparable. Scaffold-GS (Lu et al., 2024) binds Gaussians to sparse anchor points and predicts attributes on-the-fly, pruning redundancy, and improving view-adaptivity with fewer primitives at the cost of per-view MLP inference and extra book-keeping. Mip-Splatting (Yu et al., 2024) constrains the Gaussian size via a 3D smoothing prior and applies 2D mip-filtering, eliminating zoom-induced aliasing, although its rendering remains splat-bound and memory-heavy. Moreover, SuGaR (Guédon & Lepetit, 2024) adds a surface-alignment regularizer that exploits Poisson reconstruction to extract editable meshes in minutes, forming a hybrid mesh-plus-splat representation for easy editing while inheriting the single-stream color model from 3DGS. Ref-GS (Zhang et al., 2025c) further enhances 2D Gaussian Splatting by introducing deferred shading and directional factorization, effectively modeling specular highlights and geometry under near-field lighting conditions. However, these methods still struggle to handle complex lighting effects and require substantial memory to store per-point attributes.

## 2.2 Tensor Decomposition for Efficient NVS

Tensor decomposition has been widely studied in signal processing, computer vision, and machine learning (Sidiropoulos et al., 2017; Cichocki et al., 2015; Chen et al., 2022b; He et al., 2019; Zhang et al., 2020; Marquez et al., 2020; Zhang et al., 2021; Ji et al., 2019). Among them, the most commonly used is Canonical Polyadic (CP) decomposition (Carroll & Chang, 1970), which expresses a tensor as a sum of rank-one components, and Tucker decomposition (Tucker, 1966), a higher-order generalization of matrix SVD. More recently, Vector-Matrix (VM) decomposition (Chen et al., 2022a) has been introduced to improve efficiency by representing some tensor modes with matrices and others with vectors.

Tensor decomposition has been increasingly integrated into NVS frameworks for compact and structured scene encoding. For instance, TensoRF (Chen et al., 2022a) maps the 3D spatial position to an appearance feature using CP or VM decomposition. It shrinks the model size and reduces the reconstruction time while matching or outperforming the baseline quality. Tensor4D (Shao et al., 2023) extends TensoRF to dynamic scenes using hierarchically decomposed 4D spatiotemporal tensors to deliver high-fidelity reconstruction from sparse views. TensoIR (Jin et al., 2023) couples low-rank tensor factors with inverse rendering to jointly recover scene geometry, surface reflectance, and environment illumination. Moreover, Gaussian-based methods such as MiGS (Chatziagapi et al., 2024) and 3iGS (Tang & Cham, 2024) apply tensor decomposition to represent per-Gaussian features or lighting fields. MiGS (Chatziagapi et al., 2024) forms a high-order tensor as all learnable 3DGS parameters and applies CP decomposition to build a multi-identity Gaussian set that reuses geometry across subjects, reducing the memory requirement. 3iGS (Tang & Cham, 2024) replaces per-splat spherical harmonics with a factorized tensor illumination field and regresses per-Gaussian bidirectional reflectance distribution function (BRDF) features.

Nevertheless, these tensorial representations rely on classic tensor decomposition, prioritizing learning the low-frequency components and failing to capture high-frequency specular and shadow reflections under complex lighting.

## 3 Method

Unlike 3DGS, which uses SHs for modeling Gaussian color, this paper maintains 3DGS's geometric parameters—rotation, scaling, and opacity $\alpha$—and introduces a new scene color modeling strategy, as shown in Fig. 2. The scene color is decomposed into a view-independent base color and a view-dependent residual color. The residual color is predicted by a neural TR Regression model that learns view-dependent appearance features from spatial positions and viewing directions. To facilitate understanding, we first define the notations and preliminaries (Sec. 3.1), followed by the detailed introduction of our scene color learning pipeline (Sec. 3.2). Finally, we describe the optimization strategy and loss functions used during training (Sec. 3.3).

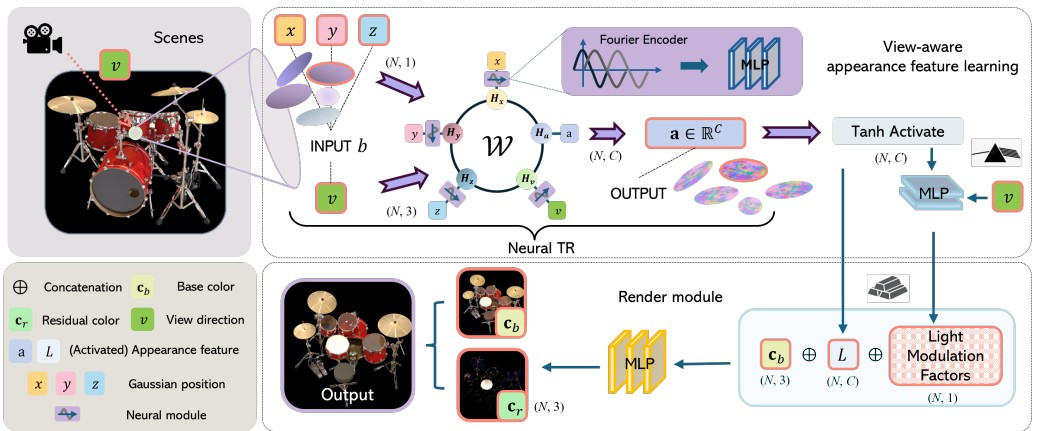

Figure 2: Overview of the proposed scene color modeling framework. The view-dependent residual color is obtained by two main stages: (i) a view-aware appearance feature $\mathbf{a} \in \mathbb{R}^C$ is constructed via a TR function regression from positions $(x, y, z)$ and viewing direction $v$ (see Sec. 3.2 for details); (ii) a render module predicts a view-dependent residual color by modulating $\mathbf{a}$ with a learned light modulation scalar, which is added to a base color $\mathbf{c}_b$ to produce the final output.

## 3.1 NOTATIONS AND PRELIMINARIES

As depicted in Fig. 3, the notations for a scalar, a vector, a matrix, and a tensor are $x$, $\mathbf{x}$, $\mathbf{X}$, and $\mathcal{X}$. Index values typically range from 1 to their uppercase counterpart, e.g., $i = 1, \cdots, I$.

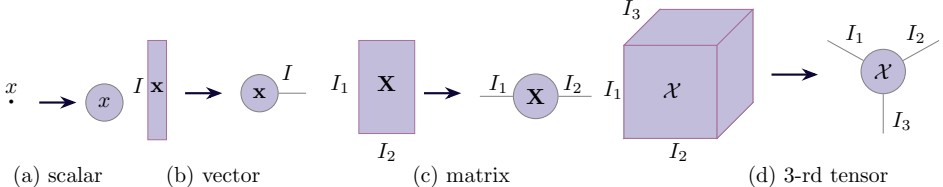

(a) scalar      (b) vector      (c) matrix      (d) 3-rd tensor

Figure 3: The graphical illustration of tensor notations.

**Definition 1.** *(**Tensor Contraction**) The contraction of two tensors, denoted $\mathcal{X} \in \mathbb{R}^{I_1 \times I_2 \times J_1 \times J_2}$ and $\mathcal{Y} \in \mathbb{R}^{J_1 \times J_2 \times L_1 \times L_2}$, entails summation over shared indices as noted in $\{J_1, J_2\}$. This process yields the result $\mathcal{Z} = \mathcal{X} \times_{\{J_1, J_2\}} \mathcal{Y} \in \mathbb{R}^{I_1 \times I_2 \times L_1 \times L_2}$, where the entries are evaluated according to $c_{i_1, i_2, l_1, l_2} = \sum_{j_1, j_2} a_{i_1, i_2, j_1, j_2} b_{j_1, j_2, l_1, l_2}$.*

**Definition 2.** *(**Tensor Ring Decomposition**) (Zhao et al., 2016) Consider a Dth-order tensor $\mathcal{X} \in \mathbb{R}^{I_1 \times \cdots \times I_D}$, with a D-tuple $[R_1, \ldots, R_d, \ldots, R_D]$ satisfying $R_{D+1} = R_1$, such that*

$$\mathcal{X}(i_1, i_2, \ldots, i_D) = \text{Trace}(\mathcal{G}_1(:, i_1, :)\mathcal{G}_2(:, i_2, :) \cdots \mathcal{G}_D(:, i_D, :)), \tag{1}$$

*where the core components are $\mathcal{G}_d \in \mathbb{R}^{R_d \times I_d \times R_{d+1}}$ for $d = 1, \ldots, D$. For ease of expression, we represent the TR decomposition of $\mathcal{X}$ as $\mathcal{X} = \mathfrak{R}(\mathcal{G}_1, \ldots, \mathcal{G}_D)$. A visual demonstration of the TR decomposition is provided in Fig. 4 (a).*

**Definition 3.** *(**Tensor Regression**) (Liu et al., 2021) Given an input tensor $\mathcal{X} \in \mathbb{R}^{P_1 \times \cdots \times P_L}$ and an output tensor $\mathcal{Y} \in \mathbb{R}^{Q_1 \times \cdots \times Q_M}$, the formulation of the tensor regression model is as follows:*

$$\mathcal{Y} = \mathcal{X} \times_{\{P_1, \cdots, P_L\}} \mathcal{W} + \mathcal{E}, \tag{2}$$

*where $\mathcal{W} \in \mathbb{R}^{P_1 \times \cdots \times P_L \times Q_1 \times \cdots \times Q_M}$ is the coefficient tensor, $\mathcal{E} \in \mathbb{R}^{Q_1 \times \cdots \times Q_M}$ is the error tensor.*

**Definition 4.** *(**Gaussian Splatting**) (Kerbl et al., 2023) 3D Gaussian Splatting for NVS rasterizes scenes represented by anisotropic 3D Gaussians. Each Gaussian is defined by a center $\boldsymbol{\mu} = (x, y, z) \in \mathbb{R}^3$, a covariance matrix $\boldsymbol{\Sigma} \in \mathbb{R}^{3 \times 3}$, and appearance attributes. The Gaussian density is given by:*

$$g(\mathbf{x} \mid \boldsymbol{\mu}, \boldsymbol{\Sigma}) = \exp\left(-\tfrac{1}{2}(\mathbf{x} - \boldsymbol{\mu})^\top \boldsymbol{\Sigma}^{-1}(\mathbf{x} - \boldsymbol{\mu})\right), \tag{3}$$

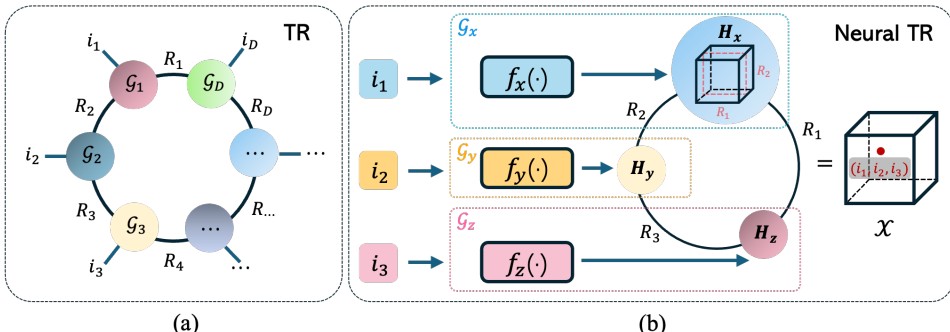

Figure 4: The graphical illustration of TR and neural TR decomposition.

with $\mathbf{\Sigma}$ decomposed as $RSS^\top R^\top$. After projection into camera space via transformation $\mathbf{J}$, the screen-space covariance becomes $\mathbf{\Sigma}' = \mathbf{JW\Sigma W}^\top \mathbf{J}^\top$. Appearance is modelled using SHs, and the final pixel color is computed via front-to-back alpha compositing:

$$\hat{C}(u) = \sum_{i \in \mathcal{N}} T_i \cdot g_i(u \mid \boldsymbol{\mu}', \mathbf{\Sigma}') \cdot \alpha_i \cdot c_i, \quad T_i = \prod_{j=1}^{i-1} \left(1 - g_j(u \mid \boldsymbol{\mu}', \mathbf{\Sigma}') \cdot \alpha_j\right). \tag{4}$$

## 3.2 SCENE COLOR LEARNING

Inspired by physically based reflectance (PBR) models, we abandon the use of SH and adopt a structured, data-driven formulation to model the color of each 3D Gaussian. Unlike traditional BRDF-based methods, which use explicit parameters such as roughness or metallicity, our model directly captures a latent reflectance structure from the learned appearance feature field. The final output color is computed via residual addition:

$$\mathbf{c}(\mathbf{v}) = \underbrace{\mathbf{c}_b}_{\text{view-independent base color}} + \underbrace{\mathbf{c}_r(\mathbf{v})}_{\text{view-dependent residual color}} \tag{5}$$

Here, the base color $\mathbf{c}_b \in \mathbb{R}^3$, representing a per-Gaussian diffuse albedo, remains invariant across viewing directions. To enable expressive modeling of view-dependent illumination effects in Gaussian-based rendering, the residual color is proposed, as shown in Fig. 2. The residual color is achieved through two steps: view-aware appearance feature learning followed by a render module.

### 3.2.1 VIEW-AWARE APPEARANCE FEATURE LEARNING

To capture the complex interactions between spatial geometry and view-dependent illumination, this work proposes a Neural TR Regression (NTRR) method to model the mapping between spatial positions $x, y, z$, viewing direction $v$, and view-dependent appearance features $\mathbf{a} \in \mathbb{R}^C$, as below:

$$\mathbf{a} = \mathcal{B}(x, y, z, v) \times_{\{X, Y, Z, V\}} \mathcal{W}, \tag{6}$$

where the 5th-order tensor $\mathcal{W} \in \mathbb{R}^{X \times Y \times Z \times V \times C}$ is the regression coefficient, which aims to map the discrete input $b = \mathcal{B}(x, y, z, v)$ to the view-dependent appearance feature $\mathbf{a}$. Note that $b$ is not a simple concatenation and each element of $b$ is processed by its corresponding TR core in $\mathcal{W}$. Here, $\mathcal{W}$ follows a neural TR decomposition as follows:

**Definition 5.** *(Neural TR Decomposition) Given a 5th-order tensor $\mathcal{X} \in \mathbb{R}^{X \times Y \times Z \times V \times C}$ with arbitrary bounded functions $f_x(\cdot) : \mathbb{X}_f \to \mathbb{R}^{R_1 \times R_2}$, $f_y(\cdot) : \mathbb{Y}_f \to \mathbb{R}^{R_2 \times R_3}$, $f_z(\cdot) : \mathbb{Z}_f \to \mathbb{R}^{R_3 \times R_4}, f_v(\cdot) : \mathbb{V}_f \to \mathbb{R}^{R_4 \times R_5}$, $f_c(\cdot) : \mathbb{C}_f \to \mathbb{R}^{R_5 \times R_1}$, the Neural TR decomposition is defined as:*

$$\mathcal{X}(x, y, z, v, c) = \text{Trace}(\mathcal{G}_x(:, x, :)\mathcal{G}_y(:, y, :)\mathcal{G}_z(:, x, :)\mathcal{G}_v(:, v, :)\mathcal{G}_c(:, c, :)). \tag{7}$$

*Here, $\mathcal{G}_x(:, x, :) = f_x(x) = \mathbf{H}_x \in \mathbb{R}^{R_1 \times R_2}$ is the neural TR core tensor, and $f_x(\cdot)$ denotes a neural module, which combines Fourier feature mappings with a learnable MLP to embed the discrete inputs into a high-dimensional continuous latent space. $\mathbf{H}_x$ serves as the neural counterpart of the TR slice. The same applies to other $\mathcal{G}(\cdot)$. Unlike conventional TR decomposition, which is a fixed parameter slice indexed by $i_d$, Neural TR generates this slice dynamically through a neural mapping. $R_d, d = 1, \cdots, 5$ extend TR rank from discrete tensors to continuous tensor functions. Fig. 4 (b) illustrates a 3rd-order tensor through neural TR.*

### 3.2.2 RENDER MODULE

The view-aware appearance feature $\mathbf{a} \in \mathbb{R}^C$ is first activated and modulated by the view direction $\mathbf{v}$. The rendering module then combines three components: the appearance feature $\mathbf{a}$, the base color $\mathbf{c}_b$, and the light modulation factors. Here, The light modulation factors act as blending weights that adjust the contribution of view-dependent appearance, modulating the strength of specular highlights. A more detailed interpretation and visualization of the factors is provided in the Appendix C. A lightweight MLP processes these components to estimate a specular residual color $\mathbf{c}_r(\mathbf{v}) \in \mathbb{R}^3$, defined as:

$$\mathbf{c}_r(\mathbf{v}) = f_{\text{MLP}}(\mathbf{a}, f_{\text{MLP}}(\mathbf{v}, \mathbf{a}), \mathbf{c}_b) \tag{8}$$

Note that compared to the original 3DGS framework, where each Gaussian independently optimizes its view-dependent behavior, our neural TR-based illumination field enables the model to capture global regularities in lighting by sharing structure across all spatial locations. In contrast to 3iGS, which relies on Vector-Matrix tensor factorization, neural TR decomposition provides higher expressive capacity by modeling high-frequency information, while maintaining compact parameterization. Moreover, by explicitly incorporating the viewing direction as an early-stage input, our method allows the illumination field to reason about view-dependent effects at the representation level, rather than relying on post-hoc shading.

## 3.3 OPTIMIZATION STRATEGY

We optimize our model using a hybrid loss that combines pixel-wise fidelity and structural similarity:

$$\mathcal{L} = \gamma \cdot [(1 - \beta) \cdot \mathcal{L}_2 + \beta \cdot \mathcal{L}_{\text{SSIM}}], \tag{9}$$

where $\gamma$ and $\beta$ are hyperparameters. The loss term $\mathcal{L}_2$ emphasizes the accuracy in pixels and is particularly effective in capturing high-frequency details, which are crucial to faithfully modeling view-dependent effects. The SSIM term promotes structural consistency between regions. We apply the scaling factor $\gamma$ to ensure that the overall loss magnitude is large enough to drive effective optimization of the TR core parameters.

In addition, we introduce a learnable visibility mask $m_n \in \mathbb{R}$ for each Gaussian (Kim et al., 2024), trained via a self-supervised gating mechanism based on a straight-through estimator (STE) (Bengio et al., 2013). The binary gating mask is computed as follows:

$$M_n = sg\left(\mathbf{1}\left[\sigma(m_n) > \epsilon\right] - \sigma(m_n)\right) + \sigma(m_n). \tag{10}$$

where $sg(\cdot)$ is the stop-gradient operator, $\sigma(\cdot)$ is the sigmoid function, and $\epsilon$ is a threshold. The mask modulates the opacity of each Gaussian: $\hat{\alpha}_n = M_n \cdot \alpha_n$.

## 4 EXPERIMENTS AND RESULTS

### 4.1 EXPERIMENTAL SETTINGS

**Evaluation Datasets and Metrics.** To comprehensively evaluate the performance of our method in different scenarios, we performed experiments on three representative datasets. The *Tanks and Temples* benchmark (Knapitsch et al., 2017) has been adopted to assess the model's ability to handle complex, large-scale outdoor scenes with intricate geometry. To evaluate performance under challenging view-dependent appearance, we further employed the *NeRF-Synthetic* (Mildenhall et al., 2021) and *Shiny-Blender* (Verbin et al., 2022) datasets. *NeRF-Synthetic* includes clean, indoor scenes with moderate specular effects, while *Shiny-Blender* features simplified object-centric scenes with strong specular highlights and glossy reflections. To ensure fair comparisons, we followed the official train-test splits provided in 3DGS (Kerbl et al., 2023) for all datasets. Quantitative evaluation was performed using PSNR, SSIM (Wang et al., 2004), and LPIPS (Zhang et al., 2018) to measure perceptual image quality, and model sizes to assess memory and storage efficiency. In addition, we also report the rendering speed (FPS) and the total training time to assess computational efficiency, as detailed in Appendix F.

**Comparison Baselines.** We compared our method with representative approaches classified in the NeRF-based and 3DGS-based frameworks. For baselines based on NeRF, we have included TensoRF (Chen et al., 2022a) and PuTT (Loeschcke et al., 2024). These approaches apply tensor decomposition to compactly represent volumetric radiance fields. Among 3DGS-based methods, we have

Table 1: Quantitative results evaluated on three datasets. Best and second-best results are highlighted in red and yellow, respectively.

| Dataset / Method | Tanks and Temples | | | | NeRF-Synthetic | | | | Shiny-Blender | | | |
|---|---|---|---|---|---|---|---|---|---|---|---|---|
| | SSIM | PSNR | LPIPS | Size (MB) | SSIM | PSNR | LPIPS | Size (MB) | SSIM | PSNR | LPIPS | Size (MB) |
| TensoRF (Chen et al., 2022a) | 0.881 | 25.53 | 0.180 | ╲ | 0.943 | 30.08 | 0.067 | ╲ | 0.916 | 26.67 | 0.123 | ╲ |
| PuTT (Loeschcke et al., 2024) | 0.873 | 25.33 | 0.199 | ╲ | 0.936 | 30.06 | 0.072 | ╲ | 0.910 | 26.19 | 0.117 | ╲ |
| 3DGS (Kerbl et al., 2023) | 0.925 | 28.92 | 0.104 | 200.47 | 0.967 | 32.90 | 0.044 | 68.35 | 0.945 | 28.87 | 0.101 | 46.43 |
| 3iGS (Tang & Cham, 2024) | 0.928 | 29.06 | 0.105 | 272.38 | 0.969 | 33.27 | 0.044 | 78.49 | 0.951 | 28.90 | 0.104 | 65.44 |
| GaussianShader (Jiang et al., 2024) | 0.926 | 27.97 | 0.107 | 229.99 | 0.972 | 32.92 | 0.043 | 79.31 | 0.943 | 28.52 | 0.103 | 53.41 |
| 3DGS-DR (Ye et al., 2024) | 0.918 | 27.44 | 0.118 | 163.53 | 0.969 | 33.05 | 0.045 | 54.28 | 0.953 | 28.89 | 0.100 | 31.57 |
| Glossy-GS (Zhang et al., 2025a) | 0.931 | 29.14 | 0.105 | 301.24 | 0.971 | 33.26 | 0.043 | 82.41 | 0.946 | 28.48 | 0.103 | 57.13 |
| LightGaussian (Fan et al., 2024) | 0.905 | 26.56 | 0.140 | 57.97 | 0.961 | 31.24 | 0.061 | 16.22 | 0.927 | 27.24 | 0.136 | 15.50 |
| **RingLight-GS (Ours)** | 0.934 | 29.47 | 0.100 | 75.91 | 0.975 | 33.61 | 0.041 | 21.9 | 0.957 | 29.01 | 0.096 | 15.95 |

considered 3iGS (Tang & Cham, 2024), which combines tensor decomposition with physics-based rendering, GaussianShader (Jiang et al., 2024), which explicitly models specular highlights through analytic shading, and 3DGS-DR (Ye et al., 2024), which introduces a deferred shading mechanism to improve specular appearance modeling. We have also included the original 3DGS (Kerbl et al., 2023) and the compressed variant LightGaussian (Fan et al., 2024) to evaluate both high-frequency detail reconstruction and model compactness.

**Implementation Details.** Our implementation is based on 3DGS (Kerbl et al., 2023) and 3iGS (Tang & Cham, 2024). All experiments were performed on a server running Ubuntu 20.04 using a single NVIDIA A100 GPU. We uniformly set the tensor ring core ranks to 16 and the appearance feature dimension $\mathbf{a}$ to 48. We adopt Fourier positional encoding with 10 frequency bands to embed spatial and view-directional inputs. The loss weighting parameters are set to $\gamma = 10^4$ and $\beta = 0.2$, as described above. The densification threshold is fixed at $1.1$. All other hyperparameters follow the default settings of the original 3DGS implementation.

## 4.2 EXPERIMENTAL RESULTS

The evaluation results for three datasets are summarized in Table 1. Compared to NeRF-based baselines (Chen et al., 2022a; Loeschcke et al., 2024), 3DGS-based methods can achieve superior rendering quality across all metrics, highlighting the advantage of point-based representations for photorealistic view synthesis. Building upon 3DGS (Kerbl et al., 2023), RingLight-GS achieves a substantial reduction in model size while enhancing visual fidelity, particularly in high-frequency regions and scenes under complex lighting conditions. Our method reduces the storage by approximately 3.1×, 2.7×, and 2.5× on the three datasets, respectively, demonstrating the efficiency of our compact directional encoding. Furthermore, RingLight-GS outperforms recent baselines that are specifically designed to model complex lighting effects. Compared to 3iGS (Tang & Cham, 2024) and GaussianShader (Jiang et al., 2024), it achieves better perceptual quality with less memory usage. Although LightGaussian (Fan et al., 2024) provides strong model size reduction, it suffers from noticeable quality degradation, particularly under complex lighting conditions.

Fig. 5 presents qualitative comparisons between RingLight-GS and other baselines. Our method can preserve sharp highlights, metallic textures, and reflective surfaces, particularly under strong lighting variations. In the Toaster and Drums scenes, RingLight-GS produces clearer contours and more accurate specular details, whereas other methods suffer from over-smoothing or blurring.

## 4.3 ABLATION STUDIES

To evaluate the effectiveness of our key design choices, we conducted ablation studies on the *Shiny-Blender* (Verbin et al., 2022) dataset. Specifically, we examined the contribution of two critical components: the neural TR-based decomposition and the explicit base color representation. The results are shown in Table 2 and Fig. 6.

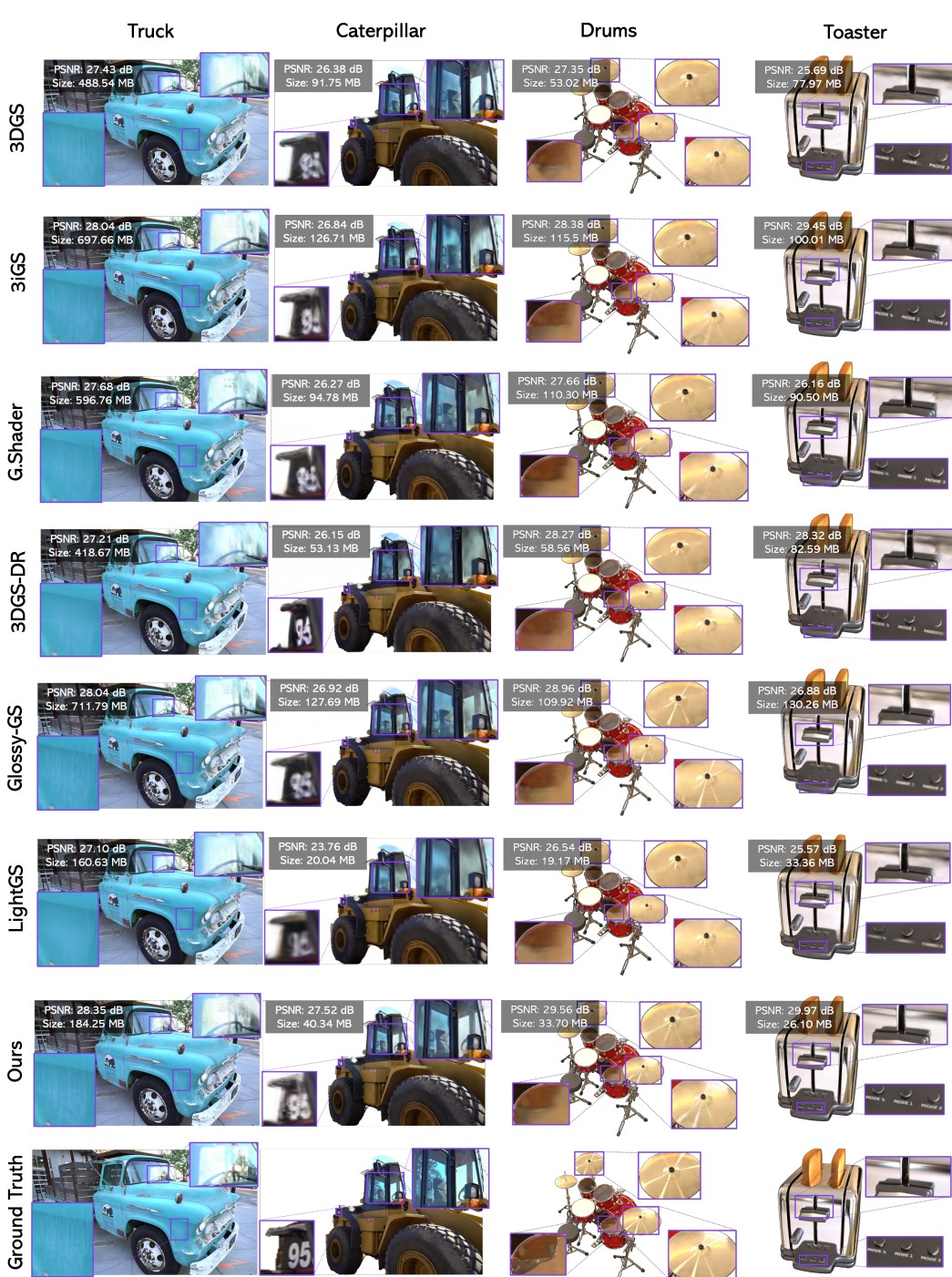

Figure 5: Qualitative comparisons on four scenes. Zoomed-in regions highlight improved detail recovery compared to prior methods.

**Effectiveness of Neural TR Module.** We evaluated the Neural TR Module through two experiments: first, by omitting the Neural component, resulting in the "w/o Neural Module" that uses the traditional TR structure; second, by replacing the TR structure with an MLP, called "w/o TR".

Fig. 6 (a) shows that without the neural module, the traditional TR structure directly processes input coordinates, limiting its capacity to model high-frequency lighting. Consequently, the outputs display over-smoothed shading and lack sharp highlights or fine detail, particularly in glossy and re-

Table 2: Ablation study on the *Shiny-Blender* dataset (Verbin et al., 2022). We report PSNR, SSIM, and LPIPS to evaluate the contribution of each component.

| Metric | Ours (Full Model) | w/o Neural Module | w/o TR | w/o Base Color |
|--------|-------------------|-------------------|--------|----------------|
| SSIM   | **0.957**         | 0.947             | 0.923  | 0.878          |
| PSNR   | **29.01**         | 27.73             | 26.13  | 22.85          |
| LPIPS  | **0.096**         | 0.121             | 0.134  | 0.173          |

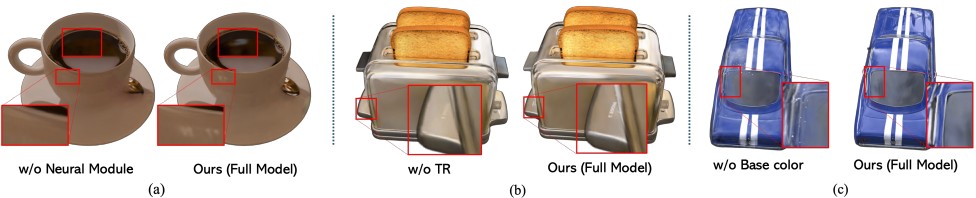

w/o Neural Module    Ours (Full Model)        w/o TR        Ours (Full Model)        w/o Base color        Ours (Full Model)

(a)                              (b)                              (c)

Figure 6: Qualitative ablation results on the *Shiny-Blender* dataset (Verbin et al., 2022). Removing the neural module leads to smooth and desaturated highlights, removing the TR decomposition and replacing it with a plain MLP leads to blurry reflections and color bleeding, while removing the base color causes unstable appearance and loss of material consistency.

flective areas. As in Fig. 6 (b), when the TR module is removed and replaced with a plain MLP that directly maps the 4D spatial-view coordinate $(x, y, z, v)$ to the appearance feature $\mathbf{a}$, the rendered outputs exhibit degraded quality. The TR decomposition provides structured spatial factorization, allowing efficient representation of view-dependent appearances with fewer parameters. Without this spatial factorization, issues like color bleeding, blurry reflections, and lack of geometric coherence occur.

**Effectiveness of Base Color.** The base color branch explicitly represents the view-independent albedo of the scene, serving as a stable foundation upon which view-dependent residuals are added. As shown in Fig. 6 (c), the removal of this component forces the lighting module to encode both static and dynamic appearance within a single representation, causing entanglement issues. In practice, this leads to color inconsistencies, exaggerated specularities, and reduced geometric coherence. The degradation is particularly evident in areas with strong material identity, where the model fails to preserve consistent surface properties across views.

## 5 CONCLUSION

We presented RingLight-GS, a compact and expressive framework for 3D Gaussian splatting that balances storage efficiency with rendering fidelity. By separating view-independent base color from view-dependent illumination and using neural TR regression for high-frequency lighting, our method offers high rendering quality under complex view-dependent effects with reduced storage costs. Experiments on synthetic and real datasets show that RingLight-GS consistently outperforms NeRF- and 3DGS-based baselines, providing sharper highlights, better material consistency, and lower perceptual error with minimal memory use. Ablation studies highlight the importance of the neural module and base color factorization for improved flexibility and stability.

Our method achieves superior rendering quality and compactness, although requiring more VRAM and longer training times compared to 3iGS (Tang & Cham, 2024) and GaussianShader (Jiang et al., 2024). The loss design emphasizes reconstruction with a factor $\gamma$, introducing a hyperparameter that warrants further impact assessment on performance and robustness. Future work aims to lower RingLight-GS's training cost and VRAM consumption through lightweight neural TR approximations, establish loss formulations for consistent illumination, and refine the prune-and-densify mechanism to improve geometric fidelity and reduce storage.

## ETHICS STATEMENT

We affirm that no anticipated ethical issues arise from this work. All data used are public or anonymized and do not involve sensitive personal information or interventions on human subjects.

## REPRODUCIBILITY STATEMENT

The source code that produces all the RingLight-GS results, including those reported here and in the main manuscript, will be publicly available on anonymous GitHub via *this link*. Detailed instructions are provided for running experiments, preprocessing data, and reproducing all figures and tables.

## LLM USAGE STATEMENT

A large language model was used only for language editing (clarity and grammar). The model did not generate any scientific content, analysis, or results. The authors take full responsibility for the scientific content of the manuscript.

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

APPENDIX: EXTENDED DISCUSSION AND EXPERIMENTAL ANALYSIS

## A    TRAINING DETAILS

Our implementation of RingLight-GS comprises two main components: (i) a neural tensor ring (TR) module to learn the view-dependent appearance and (ii) a physical render module to generate the final scene colors. We configured 30,000 iterations in total for the training phase and adopted a two-stage optimization strategy to ensure training stability. In the initial 10%, i.e., 3000 iterations, both the neural TR and the physical render modules are deactivated, allowing the model to focus exclusively on optimizing the core Gaussian parameters, including opacity $\alpha$, scale $\mathbf{S}$, rotation $\mathbf{R}$, and a view-independent base color $\mathbf{c}_b$. This warm-up phase facilitates the stabilization of geometric attributes before introducing view-dependent learning. After this stage, we activated both modules and continued end-to-end training for the remaining 27,000 iterations.

The neural TR module is implemented with a learnable MLP encoder with 128 hidden layer neurons. It takes Fourier-encoded positional information as input; The Fourier encoding includes 10 frequency bands. Configured with a constant rank of 16 across all cores, the TR representation is a compact and expressive function that maps Fourier-encoded spatial coordinates and viewing directions to a 48-dimensional appearance feature for each Gaussian. All computations are performed on a CUDA-enabled Nvidia A100 GPU.

The training process utilizes two separate Adam optimizers: one for updating the core parameters of the 3D Gaussian splats (including position, scale, rotation, opacity, and base color), the other for optimizing the parameters associated with view-dependent appearance feature learning. We adopted similar learning rate configurations as in the original 3D Gaussian Splatting benchmarks: Gaussian-related parameters were trained using an initial learning rate of 0.0025; For appearance learning, we set a learning rate of 0.0001 for the neural TR and the physical render module.

## B    MORE ABLATION STUDY

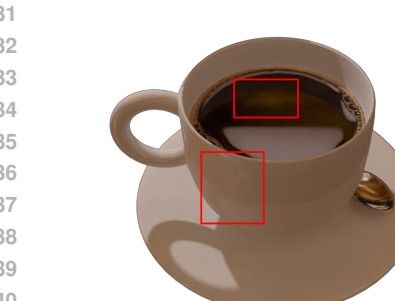 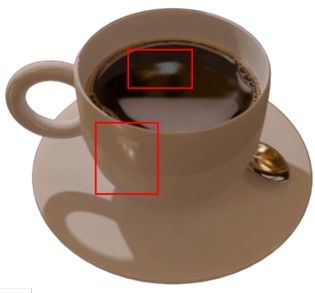

|  w/o Neural Module  |  w/o Light Modulation Factors  |  Ours (Full Model)  |

Figure 7: Qualitative results of light modulation factors ablation on the *Shiny-Blender* dataset.

Table 3: Quantitative results of light modulation factors ablation on the *Shiny-Blender* dataset.

| Metric | Ours (Full Model) | w/o Neural Module | w/o Light Modulation Factors |
|--------|-------------------|-------------------|------------------------------|
| SSIM   | **0.957**         | 0.947             | 0.955                        |
| PSNR   | **29.01**         | 27.73             | 28.94                        |
| LPIPS  | **0.096**         | 0.121             | 0.097                        |

**Effectiveness of Light Modulation Factors.** Our render module is designed to simulate view-dependent appearance by incorporating light modulation factors. To assess the importance of it, we ablated the model by simplifying the rendering function: retain only the base color and TR-derived

appearance features as input while removing the light modulation factors. As shown in Fig. 7 and Table 3, this simplification leads to degradation of the rendering quality. Specifically, outputs from this variant tend to lack fine-scale angular detail and exhibit reduced specular highlights. This indicates that although the base color and appearance feature provide a coarse approximation, modeling view-dependent cues like the light modulation factors are essential for faithfully reconstructing high-frequency effects.

## C  INTERPRETATION OF PREDICTED LIGHT MODULATION FACTOR

The light modulation factors predicted by our framework are a set of learned view-guided blending weights that modulate the contribution of view-dependent appearances. They do not represent a physically accurate BRDF parameter (as used in microfacet shading models). This set of scalars controls the angular sensitivity of the predicted color, allowing the model to dynamically interpolate between specular and diffuse effects based on the viewing direction.

To assist in interpretation, we visualize the predicted light modulation factors as grayscale overlays in Fig 8, where brighter intensities correspond to higher reflection values. As in the Toaster example, the metallic body of the toaster displays a wide range of brightness levels, with glossier areas (e.g., control knobs and polished surfaces) appearing brighter, and more diffuse components (e.g., the bread) appearing darker. Similarly, the cymbals and metal rods exhibit higher roughness in the Drums example, while the matte drum skins appear significantly darker. These patterns qualitatively align with the expected material properties and demonstrate that the learned roughness effectively captures spatially varying reflectance behavior without explicit physical supervision.

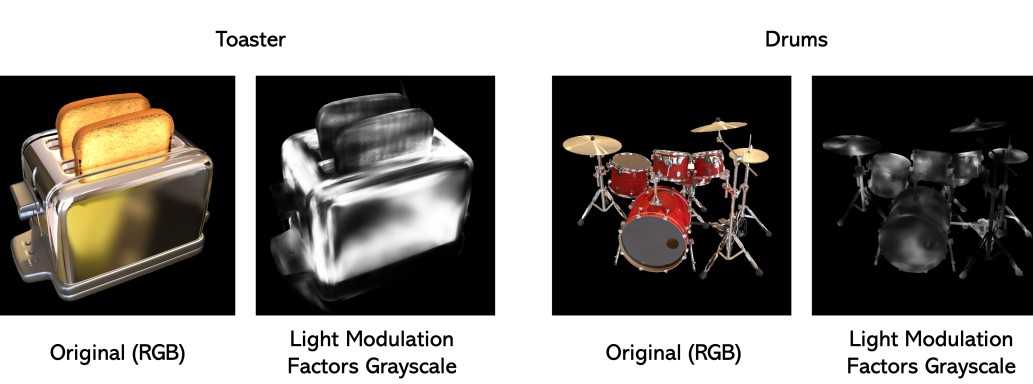

Figure 8: Visualization of the predicted light modulation factors' effect. Each pair shows the rendered RGB image (left) and its corresponding light modulation factor (right). Brighter areas denote higher values. The Toaster example exhibits both glossy and matte regions; the Drums shows clear material differentiation.

## D  COLOR DECOMPOSITION

Fig. 9 illustrates the decomposition of the scene appearance into a diffused base color and a residual view-dependent color, which are linearly combined to form the final rendered output. In the drum scene (top row), the view-dependent component recovers sharp intra-object reflections and dynamic lighting variations, demonstrating the model's ability to capture complex illumination beyond what is possible with purely view-independent features. In the train scene (bottom row), although the material is more diffuse, subtle view-dependent changes, such as shadow softening, reflectance shading, and surface interreflections, are still effectively modeled by the residual branch. Compared to standard 3D Gaussian Splatting, which does not explicitly disentangle these components, our approach produces more coherent and photorealistic results across both glossy and diffuse scenes.

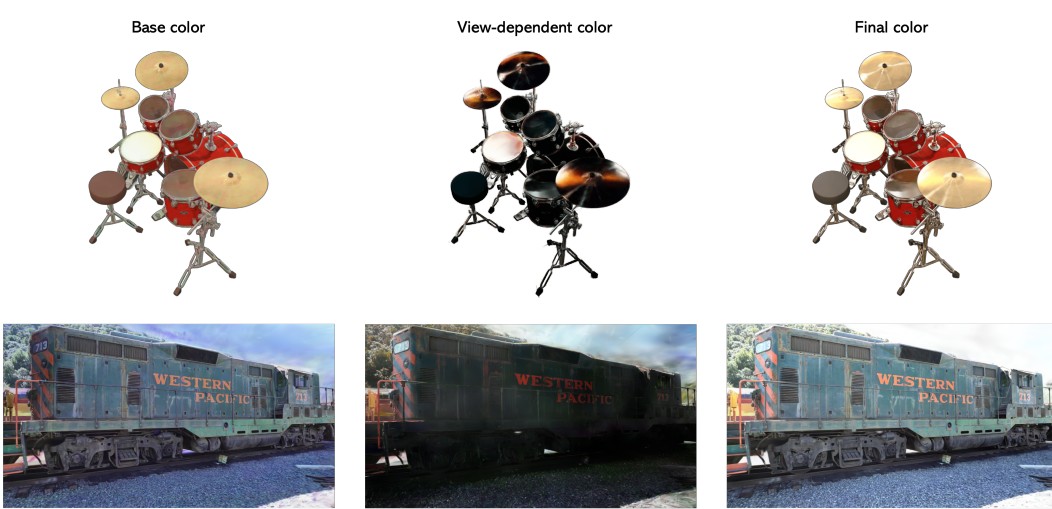

Figure 9: Decomposition of appearance into base color and view-dependent color.

# E    CHOICE OF HYPER-PARAMETERS

All hyper-parameter evaluations were conducted on the *Shiny Blender* dataset (Verbin et al., 2022)
**Ring Core Rank.** We fixed a = 48 and varied the tensor rank among {4, 8, 16, 24}. As shown in the
table below, increasing the rank improves reconstruction quality, but also increases training time and
reduces rendering speed. **Rank = 16** offers the best trade-off between quality (SSIM/PSNR/LPIPS)
and efficiency (training time and FPS).

| Config | SSIM | PSNR | LPIPS | Train Time (min) | FPS | Size (MB) |
|---|---|---|---|---|---|---|
| Rank = 4 | 0.943 | 27.71 | 0.106 | 25.01 | 161.97 | 14.16 |
| Rank = 8 | 0.955 | 28.94 | 0.096 | 26.86 | 124.67 | 14.19 |
| **Rank = 16** | **0.957** | **29.01** | **0.095** | **29.30** | **109.56** | **14.20** |
| Rank = 24 | 0.956 | 29.10 | 0.094 | 37.24 | 64.28 | 14.20 |

**Latent Appearance Dimension.** With the rank fixed at 16, we evaluated different values of a $\in$
{12, 24, 48, 60}. As presented in the following, **a = 48** provides strong reconstruction performance
with minimal overhead. Larger values (e.g., a = 60) yielded only marginal gains but incurred longer
training and slower inference.

| Config | SSIM | PSNR | LPIPS | Train Time (min) | FPS | Size (MB) |
|---|---|---|---|---|---|---|
| a = 12 | 0.946 | 28.42 | 0.100 | 29.28 | 114.96 | 14.18 |
| a = 24 | 0.951 | 28.93 | 0.096 | 29.75 | 112.76 | 14.19 |
| **a = 48** | **0.957** | **29.01** | **0.095** | **29.30** | **109.56** | **14.20** |
| a = 60 | 0.958 | 29.20 | 0.097 | 31.55 | 94.93 | 14.22 |

**MLP architecture and the loss scaling parameter** ($\gamma$). We empirically introduced a loss scaling
factor $\gamma = 10,000$ to prevent vanishing gradients during training. For the MLP architecture, we
employed a compact and effective design: a 3-layer fully connected network with a 128-neuron
hidden layer and ReLU activations. This MLP configuration follows established practices in prior
tensor representation works such as Luo et al. (2024).

**Parameter** $\beta$. The parameter $\beta$ was directly adopted from the original 3D Gaussian Splatting im-
plementation.

## F    RENDERING EFFICIENCY AND TRAINING TIME

To further assess the efficiency of our method, we compare the rendering speed (FPS) and total training time. All experiments were conducted on the same computing platform with a single NVIDIA A100 GPU. The results are summarized in Tables 4 and 5.

**Rendering Performance.**    As shown in Table 4, our method consistently achieves real-time rendering speed.

Table 4: Rendering speed (FPS) across datasets.

| Method | NeRF-Synthetic | Tanks and Temples | Shiny-Blender |
|---|---|---|---|
| 3DGS | 108.32 | 83.44 | 141.58 |
| 3iGS | 104.01 | 76.69 | 128.73 |
| GaussianShader | 107.61 | 79.00 | 131.24 |
| 3DGS-DR | 112.34 | 86.24 | 153.87 |
| Glossy-GS | 115.48 | 92.46 | 167.92 |
| LightGaussian | 166.29 | 139.42 | 218.64 |
| **RingLight-GS (Ours)** | **95.18** | **66.10** | **109.56** |
| TensoRF | 1.82 | 1.25 | 2.07 |
| PuTT | 1.49 | 0.90 | 1.74 |

**Training Time.**    As in Table 5, our method requires modest training time compared to 3DGS-style methods and remains significantly more efficient than NeRF-based pipelines.

Table 5: Training time (minutes) across datasets.

| Method | NeRF-Synthetic | Tanks and Temples | Shiny-Blender |
|---|---|---|---|
| 3DGS | 9.48 | 26.77 | 14.75 |
| 3iGS | 15.47 | 30.14 | 17.21 |
| GaussianShader | 11.36 | 27.08 | 15.42 |
| 3DGS-DR | 7.74 | 20.43 | 14.28 |
| Glossy-GS | 20.01 | 40.82 | 26.56 |
| LightGaussian | 6.26 | 19.50 | 14.61 |
| **RingLight-GS (Ours)** | **19.30** | **44.21** | **29.30** |
| TensoRF | 53.68 | 154.24 | 91.24 |
| PuTT | 126.43 | 219.54 | 156.54 |

Although RingLight-GS incurs slightly higher training cost than 3DGS, it maintains real-time rendering performance and offers significantly improved rendering quality and compact model size.

## G    FURTHER COMPARISONS

To further validate the effectiveness of RingLight-GS, we have conducted additional qualitative comparisons against four state-of-the-art baselines: 3iGS, Gaussian Shader, glossy-GS and 3DGS-DR. We selected a diverse set of scenes that include diffuse and specular materials, challenging lighting conditions, and complex geometry. Our method consistently produces sharper details, more accurate shading, and improved view-dependent effects compared to the baselines. These results further demonstrate the benefits of our neural TR representation and render module in capturing realistic appearance under novel views.

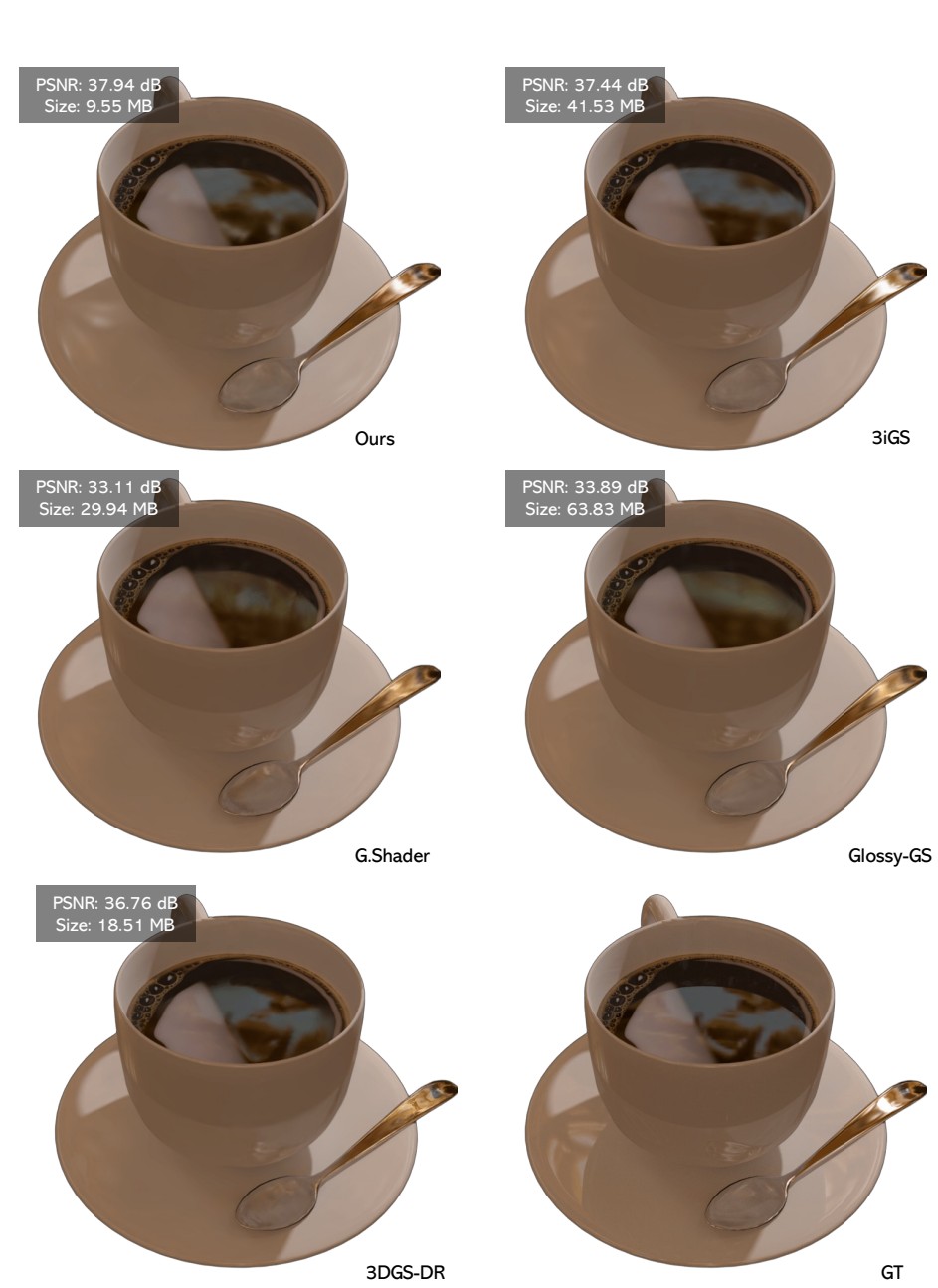

Figure 10: Comparison of coffee scene.

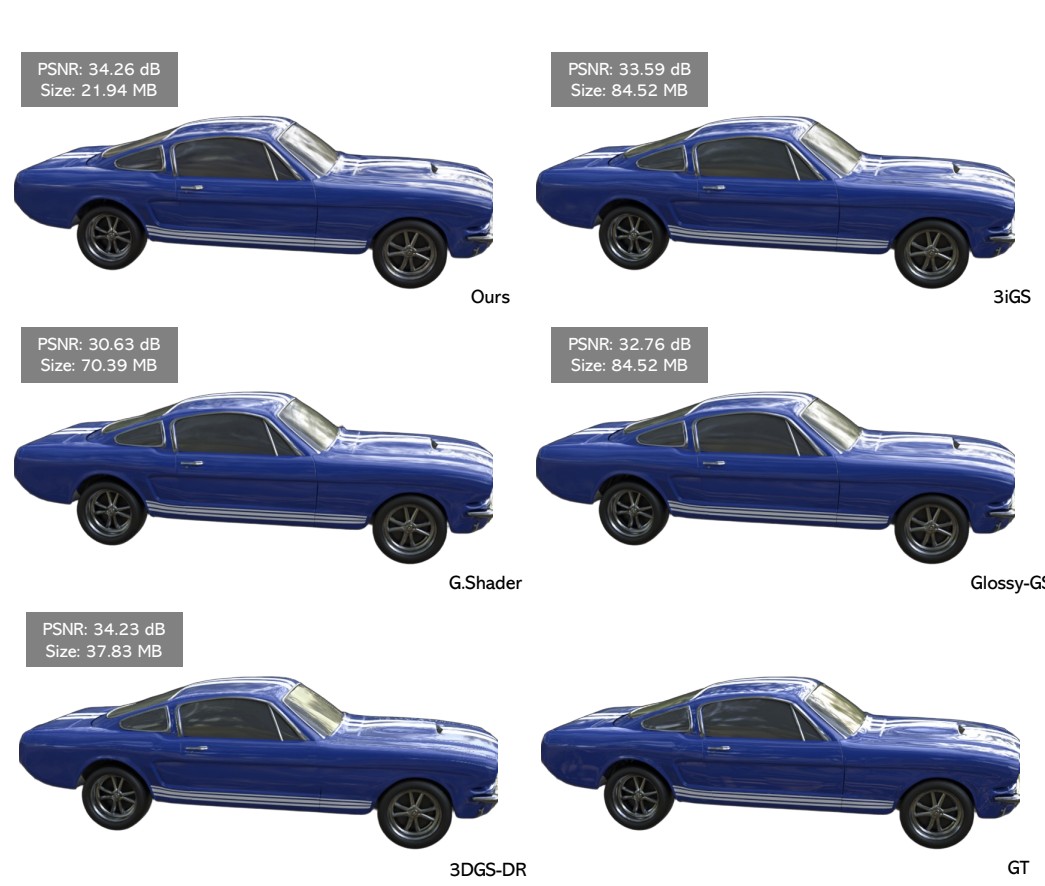

PSNR: 34.26 dB
Size: 21.94 MB

Ours

PSNR: 33.59 dB
Size: 84.52 MB

3iGS

PSNR: 30.63 dB
Size: 70.39 MB

G.Shader

PSNR: 32.76 dB
Size: 84.52 MB

Glossy-GS

PSNR: 34.23 dB
Size: 37.83 MB

3DGS-DR

GT

Figure 11: Comparison of car scene.

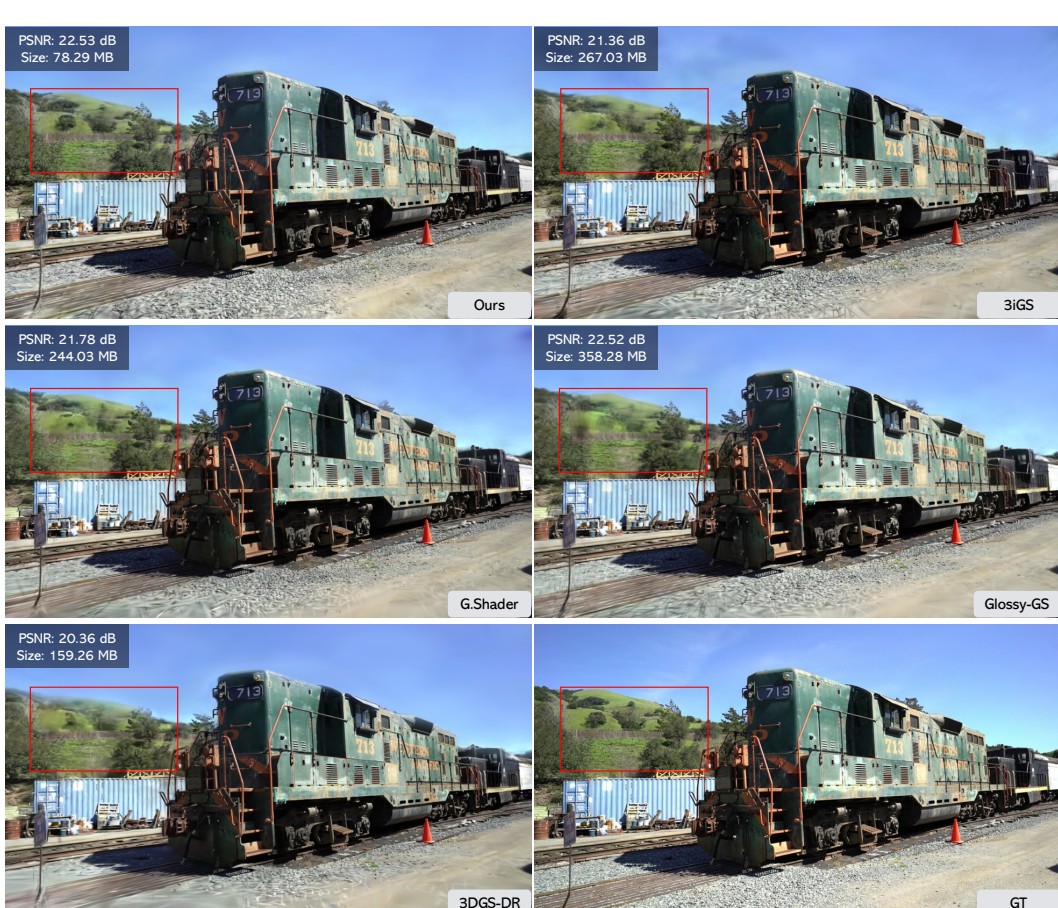

Figure 12: Comparison of train scene.

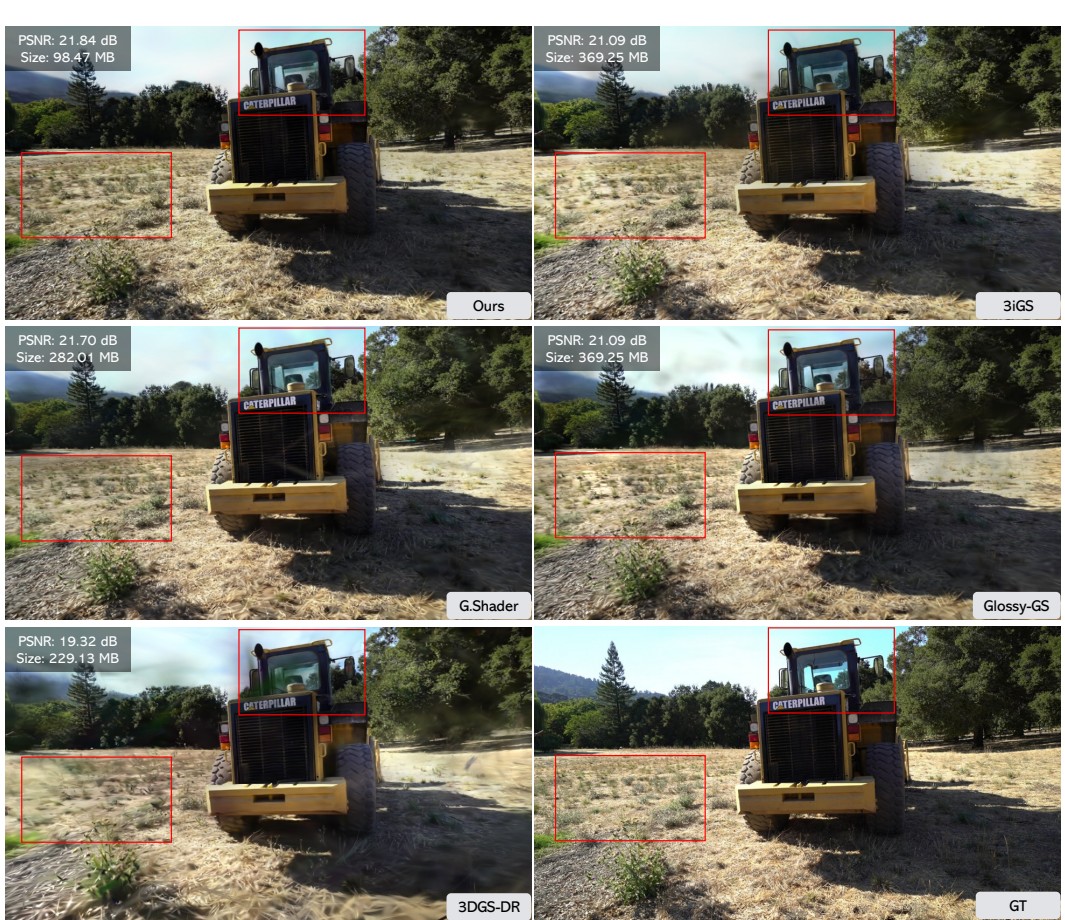

Figure 13: Comparison of caterpillar scene with background.

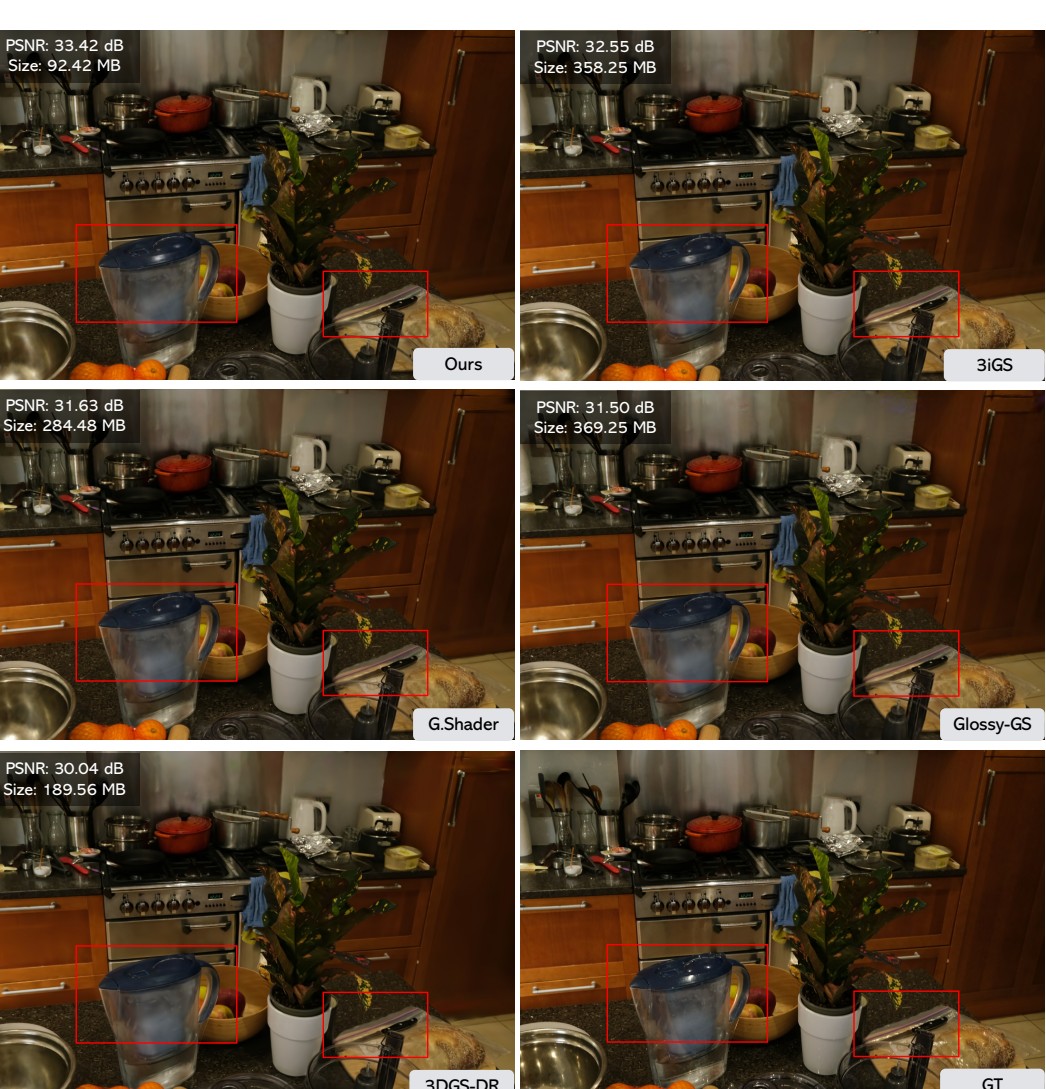

Figure 14: Comparison of counter scene.

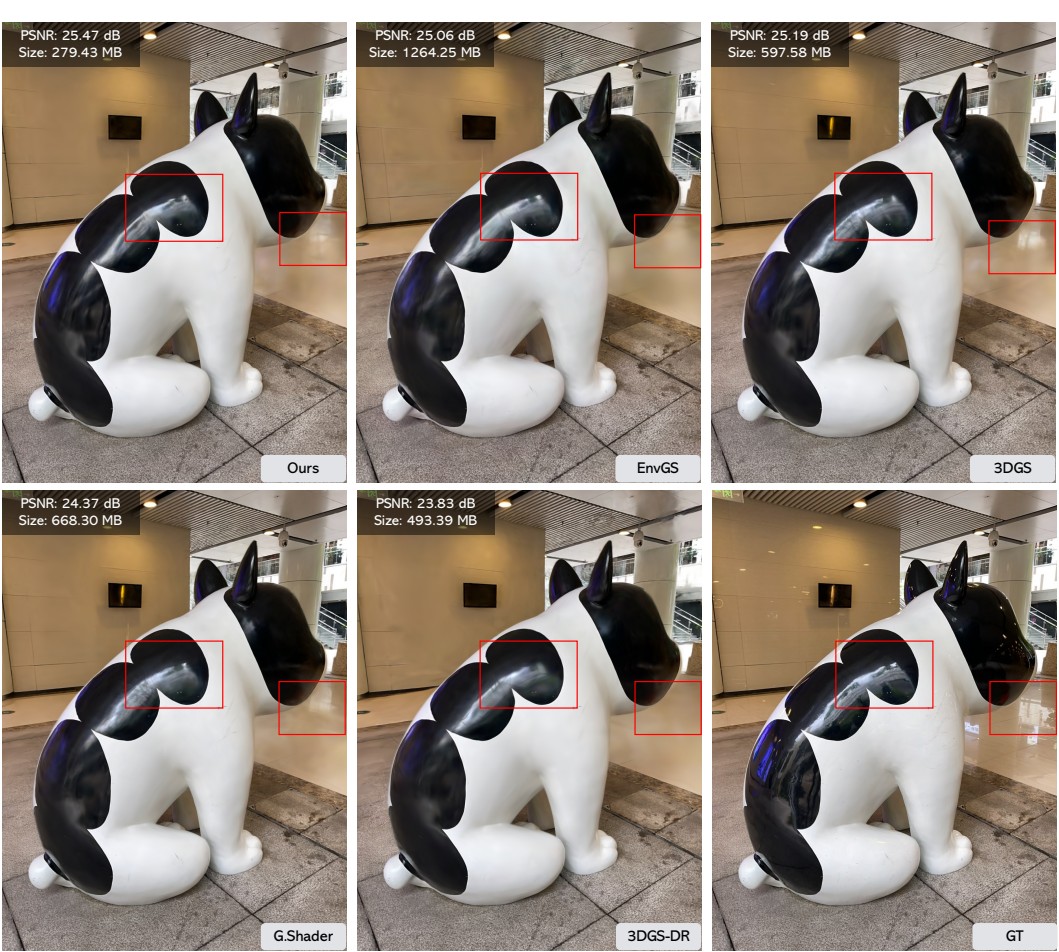

Figure 15: Comparison of dog scene.

Ours                                                 Ground Truth

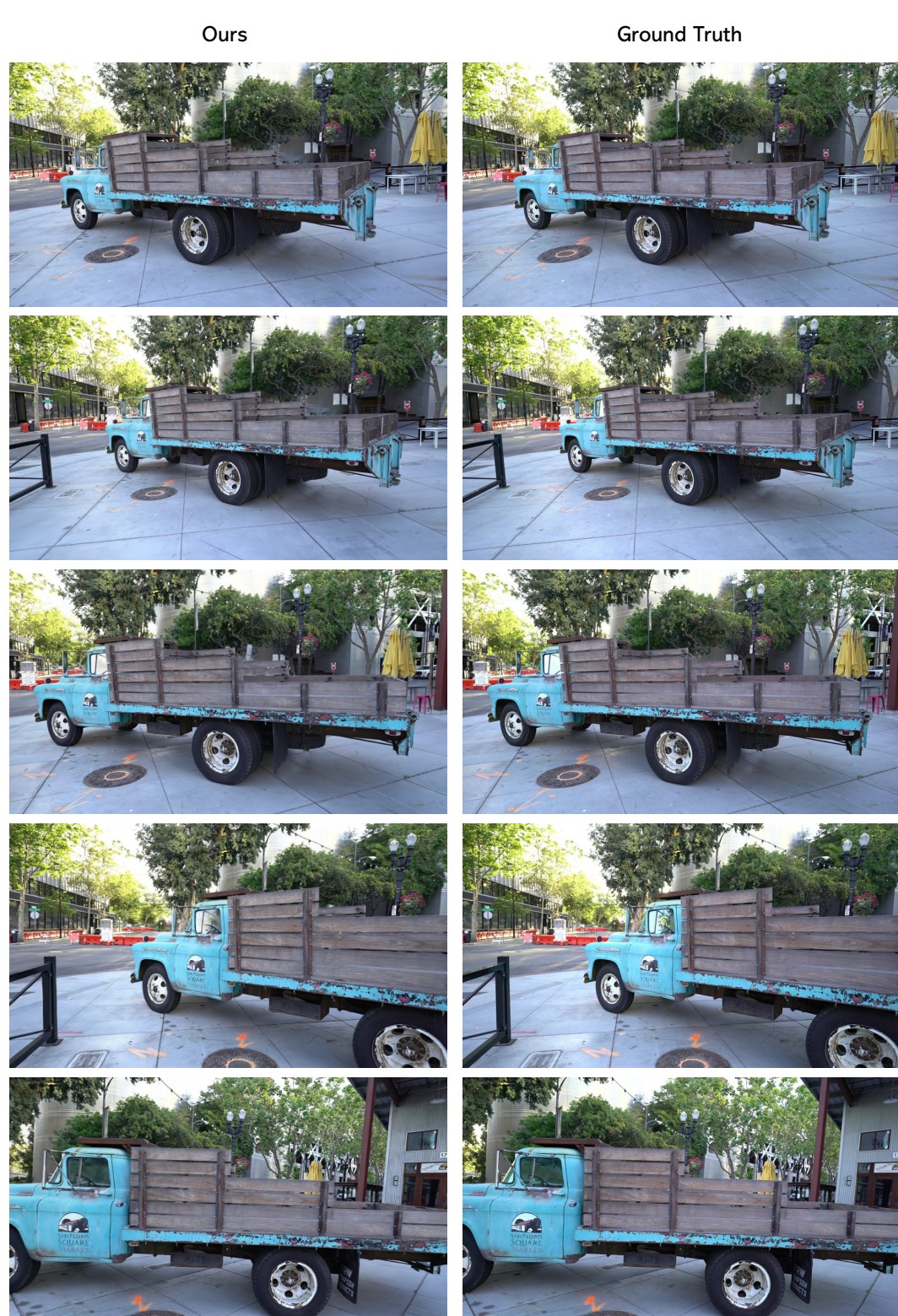

Figure 16: Qualitative demonstration of muti-view consistency.

Figure 17: Comparison with Scaffold-GS on truck scene.

