# OpenReview forum: "RingLight-GS: A compact and expressive framework for modeling scene color in 3DGS"
_ICLR.cc/2026/Conference — ICLR 2026 Conference Withdrawn Submission_

### Official Review · Reviewer_DfdE · 2025-10-27

**Soundness:** 3
**Presentation:** 3
**Contribution:** 3
**Rating:** 6
**Confidence:** 4

**Summary:**

This paper introduces RingLight-GS, a compact and expressive framework for modeling scene color in 3D Gaussian Splatting (3DGS) that aims to reduce storage costs while improving rendering quality, especially under complex lighting. The core contribution is the separation of scene color into a view-independent base color and a view-dependent residual color. This residual color, which captures high-frequency details like specular highlights, is modeled using a neural tensor ring regression. This approach allows for sharper highlights, better material consistency, and lower perceptual error with minimal memory overhead compared to previous methods.

**Strengths:**

Originality: The paper presents a original approach by integrating existing ideas into a novel 3D Gaussian Splatting framework. Its main contribution lies in modeling view-dependent residual colors using a neural tensor ring (TR) regression, combining tensor decomposition for parameter efficiency with implicit neural representations. The network dynamically generates tensor cores from continuous inputs (position and view direction), enabling a compact, continuous, and highly expressive appearance function that captures high-frequency details beyond prior tensor- or SH-based methods. This design demonstrates both technical novelty and practical effectiveness in enhancing appearance modeling in 3DGS.

Quality: The paper presents solid technical work, supported by a rigorous and comprehensive experimental evaluation. The authors validate their method on three diverse datasets—Tanks and Temples, NeRF-Synthetic, and Shiny-Blender—testing performance on complex geometry, clean indoor scenes, and challenging specular reflections, respectively. Comparisons with a wide range of NeRF- and 3DGS-based baselines are thorough and fair.

Clarity: The paper is clearly structured and written. It follows a logical progression from motivation and background to the proposed method and evaluation. Figures are used effectively: Figure 1 summarizes the method's performance against baselines, Figure 2 shows the architectural pipeline, and later figures illustrate the underlying tensor concepts. The methodology section first defines notations and preliminaries, then explains the core model components, making the technical details more accessible.

Significance: This work addresses a key challenge in 3D Gaussian Splatting (3DGS)—accurate and efficient modeling of complex, view-dependent lighting—by introducing a neural tensor representation that captures high-frequency details such as specular highlights. The method improves rendering fidelity while substantially reducing memory footprint compared to high-order Spherical Harmonics, making 3DGS more compact and practical. These improvements are relevant for applications with memory or bandwidth constraints, and the neural tensor approach may inform future research on compact representations of other complex scene properties.

**Weaknesses:**

-1. In the Optimization Strategy section, the authors introduce γ and β as hyperparameters. However, only the value of γ (set to 10,000) is briefly mentioned, without further discussion. It would strengthen the paper’s rigor if the authors could include an analysis—either in the ablation study or in the appendix—showing how different choices of these hyperparameters (e.g., γ, β, etc.) affect the results.

-2. The method separates color into a view-independent "base color" (diffuse albedo) and a view-dependent "residual." However, this decomposition is learned without direct physical supervision. The paper lacks multi-view renderings of the same scene, making it unclear whether RINGLIGHT-GS can maintain high-frequency detail consistency across views. Including such results would clarify this capability.

**Questions:**

-1. RINGLIGHT-GS leverages the TR function to compute appearance features and subsequently derive the residual color, enabling the reconstruction of high-frequency details such as specular highlights. Have the authors verified whether RINGLIGHT-GS is also effective in reconstructing diffuse surfaces?

-2. The results in Table 5 indicate that RingLight-GS requires a significantly longer training time (roughly 2-3x) compared to the original 3DGS. This represents a substantial trade-off for the achieved compactness and quality. Could you please elaborate on the primary computational bottlenecks introduced by your method?

---

> ### Author Response · Authors · 2025-11-21
> **Response to Weaknesses Pointed Out by Reviewer DfdE**
>
> **[W1] Hyperparameter settings**
>
> We thank the reviewer for pointing this out. We would like to first note that Appendix E provides the rationale for selecting various hyperparameters, along with supporting experiments.
>
> To further elaborate on the disclosed two hyperparameters. $\gamma$ is mainly used to scale the appearance loss and is empirically set to 10,000 across all experiments. It ensures that the Neural TR module receives sufficiently strong gradients during training, and without it, the neural regression network tends to suffer from vanishing gradients. In our empirical evaluation, we found that performance can stabilize across all scenes when $\gamma$ is sufficiently large (5,000 < $\gamma$ < 15000). As for $\beta$, it is a regularization weight inherited directly from the original 3DGS formulation, and we did not modify it in our method (i.e., it is a standardized constant for grounded comparison rather than a hyperparameter of our design). Therefore, we provide no extra analysis on $\beta$. We believe this further elaboration clears out the reviewer’s confusion.
>
> **[W2] Stable View-Dependent Decomposition Across Materials and Viewpoints**
>
> We understand the reviewer’s concern about whether the learned decomposition between base and view-dependent color is physically meaningful and stable across views. To clarify, our method is a data-driven method that does not rely on explicit physical modelling. The advantage of this methodology is to reduce the cost of setting up an accurate physical prior, and rely only on the dataset to train a high-performing model. However, we do acknowledge that the quality of datasets influence the model’s effectiveness, just like all existing data-driven designs.
>
> To empirically evaluate our proposal’s effectiveness, we would like to refer  to Appendix D, which illustrates how our model consistently separates view-independent base color and residual view-dependent components across both glossy and diffuse materials. As shown in Figure 9, the residual branch effectively captures sharp specularities in glossy objects (e.g., drums), while producing subtler reflectance shading in diffuse scenes (e.g., train), suggesting stable behavior across different material types without direct physical supervision. Note that the datasets we used are established open benchmarks, and we have inherited the common configurations where applicable for fair comparison and to avoid cherry-picking.
> In addition, our training framework is explicitly structured to optimize the base and residual components separately, using two independent Adam optimizers. As noted in Appendix A, one optimizer is dedicated to 3D Gaussian parameters (including base color), while the other updates the Neural TR module responsible for residual prediction. This separation ensures that both components are independently learnable and not entangled in optimization, which helps maintain multi-view consistency in practice.
>
> Finally, we have included multi-view renderings of the same scene in our Appendix G Figure 16 to provide further visual evidence of consistency under varying viewpoints. These results demonstrate that our learned residual color remains spatially and angularly coherent, even in the absence of explicit physical supervision, supporting the model’s capability to stably model high-frequency appearance variation across views.

---

> ### Author Response · Authors · 2025-11-21
> **Response to Questions Raised by Reviewer DfdE**
>
> **[Q1] Modeling View-Dependent Effects in Both Glossy and Diffuse Surfaces**
>
> We thank the reviewer for the thoughtful question. While specular highlights are indeed the most visually salient examples of high-frequency view-dependent effects, they are not the only form of such variation. Our framework models any appearance change correlated with view direction, including subtle reflectance shifts and irradiance variations common in diffuse materials, as part of the residual component. This is particularly evident in the “train” scene in Appendix D, where the material is primarily diffuse. The model still successfully separates the stable albedo from view-dependent cues such as shadow softening, interreflections, and grazing-angle shading.
>
> In addition, our benchmark includes real-captured outdoor scenes from the Tanks and Temples dataset, which exhibit natural lighting and predominantly diffuse materials. The decomposition in these scenes remains stable and visually coherent, further confirming that RingLight-GS generalizes well to diffuse surfaces and is not limited to glossy or specular objects.
>
> **[Q2] Training-Time Overhead**
>
> We acknowledge the reviewer’s observation that RingLight-GS incurs higher training costs than standard 3DGS. The primary source of this overhead is the Neural TR module, which dynamically generates high-dimensional tensor-ring cores from continuous spatial and angular embeddings via multiple neural branches with Fourier encoding. Since each Gaussian participates in this contraction-based feature computation, the process introduces additional training overhead compared to fixed SH-based representations.
>
> However, we need to note that this cost primarily applies to the training phase. For inference, the rendering remains efficient, with only a small additional overhead compared to 3DGS, and still significantly faster than NeRF-based approaches. Moreover, this trade-off during training allows us to achieve substantially better rendering quality and a much more compact model representation, particularly in challenging lighting conditions. Please refer to Appendix F for more numerical details.

---

### Official Review · Reviewer_wS37 · 2025-10-27

**Soundness:** 2
**Presentation:** 2
**Contribution:** 1
**Rating:** 2
**Confidence:** 5

**Summary:**

This paper proposes RingLight-GS, a 3DGS method that improves the color representation from SH coefficients with a neural tensor ring regression model. It decomposes each Gaussian’s color into a base color and a view-dependent residual color. The residual color is predicted via a neural TR module that takes spatial positions and view directions as input. The authors claim this design provides higher-quality, compact rendering under complex lighting conditions. Experiments show small performance improvements and storage reduction.

**Strengths:**

- The paper is easy to follow.

**Weaknesses:**

- The main claimed contribution, separating scene color into a view-independent base color and a view-dependent residual color, is a common idea and can barely be considered an original contribution. Exactly, it's just the basic rationale for the SH-based color representation that is natively adopted in 3DGS, where the degree 0 of the SH is actually the view-independent base color, and the high degrees are for the view-dependent residual color. To predict the view-dependent color more precisely, some previous works like EnvGS [1] have been proposed with the same idea, which are ignored in this work. In comparison, this work just represents a less accurate solution with weaker significance.

- 1. The introduced Neural Tensor Ring Regression is essentially not different from NeRF-style MLP with Fourier positional encoding. The TR structure merely rearranges the computation into a multilinear contraction of MLP outputs but does not change the representational capacity or introduce a meaningful inductive bias. Similar tensor decompositions have already been applied to plenty of NeRF and also 3DGS previous works. The paper does not explain why the TR form is advantageous and non-trivial.

  2. Empirical, no sufficient justification for using Tensor Ring. The only ablation (“w/o TR”) is inconclusive, considering it's unclear about the parameter count and computational cost, or the hyperparameter adjustments. From the experimental results, the paper does not show why the TR module captures high-frequency, view-dependent effects better than existing SH, MLP-based or other tensor decomposition models. No in-depth study is provided.

- Reported performance gains over 3DGS and 3iGS are small. Moreover, the comparison does not include the SOTA related methods like EnvGS for complex lighting condition, Scaffold-GS [2] equipped with MLP for better quality and compression, and other SOTA compression 3DGS works released in 2025.

[1] Xie, Tao, et al. "Envgs: Modeling view-dependent appearance with environment gaussian." Proceedings of the Computer Vision and Pattern Recognition Conference. 2025.

[2] Lu, Tao, et al. "Scaffold-gs: Structured 3d gaussians for view-adaptive rendering." Proceedings of the IEEE/CVF Conference on Computer Vision and Pattern Recognition. 2024.

**Questions:**

See the weaknesses. Overall, the method is mostly an engineering by combining existing ideas without offering new theoretical insights or a clear performance breakthrough.

---

> ### Author Response · Authors · 2025-11-21
> **Response to Reviewer wS37, part 1**
>
> **[W1] On the Potential Weak Contribution of “Base + Residual” color separation and Relation to SH / EnvGS**
>
> We acknowledge that decomposing appearance into a view-independent and a view-dependent term is an established idea, and we do not claim that this high-level concept is our novelty. Unfortunately, the reviewer misstates what we actually claim as a contribution, which is how this decomposition is implemented inside the 3DGS framework:
>
>  - We keep a per-Gaussian base color $c_b$ that explicitly plays the role of albedo (view-independent).
>  - We model the view-dependent residual $c_r(x,y,z,v)$ via a shared continuous Neural TR illumination field defined over $(x,y,z,v)$, followed by a small MLP and a light-modulation factor.
> - This residual field is shared across all Gaussians, not stored as independent per-point parameters.
>
> Therefore, this is not equivalent to SH in 3DGS:
> - SH coefficients are just a vector of trainable parameters per Gaussian. There is no enforced interpretation that degree-0 encodes “base” and higher degrees encode “residual”;
> - No mechanism to share directional structure across Gaussians;
> - No factorization across spatial and view modes.
>
> In practice, SH behaves as a dense, discrete lookup table: each Gaussian carries its own local angular basis, with no global structure. In contrast, our model forces the structure $c(v) = c_b + f_{\text{TR}}(x,y,z,v),$
> where $f_{\text{TR}}$ is: (i) a shared function across the whole scene and (ii) factorized across $(x,y,z,v)$ (see Sec. 3.2).
> This design, combined with the explicit base color, is what allows us to remove most per-Gaussian SH storage while improving view-dependent behavior.
>
> As for EnvGS, it is not the same idea either. EnvGS introduces a second set of environment Gaussians and uses a ray-tracing Gaussian renderer (based on OptiX) that casts reflection rays into an environment field using RT cores.
> Their goal is explicit environment reflection modeling with an additional Gaussian field and a dedicated ray-tracing backend.
> Our goal is different w.r.t. the following:
> - We stay within the standard 3DGS rasterization pipeline;
> - We did not introduce a second field or the usage of RT;
> - We focus on a compact, shared illumination representation that replaces SH and works on standard NVS benchmarks without requiring environment geometry or RT hardware.
>
> The two approaches are complementary: in principle, our proposal could be used as the base field in an EnvGS-style system.
>
> Finally, the claim that our method is a “less accurate solution” is contradicted by the results we report. Across all three datasets:
> We obtain better metrics than 3DGS while reducing the model size by roughly 2-to-3 times, because SH is the dominant storage term in 3DGS. On specular scenes, our qualitative results show sharper, more stable highlights than SH-based 3DGS and similar approaches. This obvious improvement seems to be largely overlooked by the reviewer, and we respectively disagree with the reviewer's statement.
>
> We will make it explicit in the revised text that the general “base + residual” concept is not claimed as our novelty; the contribution lies in the specific TR-based residual field shared across the scene, along with the explicit base color, and the resulting improvement in storage and quality.
>
> **[W2] On Neural Tensor Ring vs. NeRF-style MLP and Other Tensor Decompositions**
>
> The statement that our Neural Tensor Ring is “essentially not different from a NeRF-style MLP” and has “no meaningful inductive bias” is inaccurate. A standard NeRF-style MLP takes concatenated $(x,y,z,v)$ with positional encoding and learns an arbitrary 4D function.
>
> Our Neural Tensor Ring instead decomposes a coefficient tensor
> $\mathcal{W}(x,y,z,v)$ into mode-specific cores $\mathcal{G}x(x),\mathcal{G}y(y),\mathcal{G}z(z),\mathcal{G}v(v)$ and combines them via a multi-linear ring contraction. This enforces a low-rank, multi-linear structure across spatial and directional modes and explicitly shares the illumination structure across all Gaussians. That is a concrete inductive bias that a fully connected MLP lacks.
>
> We also tested this directly already in our ablation: the “w/o TR” variant replaces the TR illumination field with a NeRF-style MLP over
> $(x,y,z,v)$ Under a comparable parameter budget, this variant shows a large performance drop, indicating the TR structure is not a cosmetic reparameterization.
>
> Regarding other tensor decompositions (e.g., TensoRF-type methods), they typically operate on discrete voxel grids: the scene is voxelized and tensor factors are indexed by grid coordinates, which increases storage and restricts the representation to discrete samples. Our Neural TR instead takes continuous coordinates as inputs to small neural modules parameterizing each core. This avoids voxel grids, keeps storage footprint low, and integrates naturally into the 3DGS pipeline as a continuous component.
>
> We have now further clarified this aspect in Sec. 3.2.1.

---

> ### Author Response · Authors · 2025-11-21
> **Response to Reviewer wS37, part 2**
>
> **[W3] Empirical Justification and “Inconclusive Ablation"**
>
> We respectively disagree that our ablation is “inconclusive”. It directly addresses the question “why neural TR instead of a NeRF-style MLP?” by replacing our neural TR with a vanilla MLP that shows substantial degradation across all metrics and visual quality. Moreover, our “w/o base color” ablation further shows that forcing the TR field to encode both albedo and lighting severely degrades performance, confirming that the explicit base/residual split + neural TR is a meaningful design, not a cosmetic one.
>
> In addition to these analyses, we further include additional tensor decomposition baselines to strengthen the empirical justification. Specifically, we compare neural TR against CP decomposition and VM decomposition (CP: rank = 16; VM: grid resolution = 64³, appearance components = 16).
>
> | Method             | SSIM  | PSNR  | LPIPS |
> |--------------------|-------|-------|-------|
> | **Neural TR (Ours)** | **0.957** | **29.01** | **0.096** |
> | CP Decomposition   | 0.943 | 28.21 | 0.113 |
> | VM Decomposition   | 0.951 | 28.46 | 0.102 |
>
> **[W4] Small Performance Gains and Missing Baselines (EnvGS, Scaffold-GS)**
>
> Our goal is not to introduce a new 3D representation, but to improve the color/illumination representation of 3DGS. In this context, the contribution is:
>  - Compression: SH dominates 3DGS storage; our TR-based illumination field reduces overall model size by 2-to-3 times.
>  - Quality: We improve PSNR/SSIM/LPIPS on all standard datasets and visibly improve specular appearance on challenging scenes.
>
> For our proposal, which can essentially become a drop-in module, these are meaningful gains, consistent with how incremental improvements in the 3DGS and NeRF literature are usually evaluated.
>
> As noted above, EnvGS is not directly comparable as a baseline: it introduces environment Gaussians and a ray-tracing renderer specialized for explicit reflection modeling and RT hardware. Our method remains in the standard 3DGS rasterization framework and targets general NVS. However, we do appreciate the reviewer's suggestion and will cite EnvGS and clarify this difference. Further, we also conduct a direct comparison on their proposed dataset against EnvGS. A visual comparison is provided in Appendix G, Figure 15.
>
> | Sence Audi       | SSIM  | PSNR  | LPIPS | Size (Mb) |
> |--------------|-------|--------|--------|-----------|
> | **RingLight-GS (Ours)** | 0.861 | 27.52 | 0.197 | 197.94    |
> | EnvGS        | 0.862 | 27.55 | 0.193 | 673.34    |
>
> | Sence Dog       | SSIM  | PSNR  | LPIPS | Size (Mb) |
> |--------------|-------|--------|--------|-----------|
> | **RingLight-GS (Ours)** | 0.871 | 24.82 | 0.213 | 279.43    |
> | EnvGS        | 0.869 | 24.80 | 0.213 | 1264.25   |
>
> Methods like Scaffold-GS focus on geometry/attribute compression (anchors, shared features, etc.), which is largely orthogonal to our illumination representation implementation. Our TR-based color model can, in principle, be combined with such approaches.
>
> **[Q1] The “Mostly Engineering, No Clear Breakthrough” Statement**
>
>  In a nutshell, we do not claim a theoretical breakthrough; this is an application-oriented paper, and within that scope, we believe the combination of:
>  - A continuous, shared Neural Tensor Ring illumination field over $(x,y,z,v)$
>  - An explicit base/residual decomposition tailored to 3DGS, with 2-to-3-times storage reduction with improved or matched quality on standard benchmarks,
>
> constitutes a non-trivial and practically relevant contribution, supported by targeted ablations. We will make the scope and positioning clearer in the revision by refactoring our contribution statement.

---

> ### Comment · Reviewer_wS37 · 2025-11-21
>
> Thanks for the rebuttal. However, my concerns haven't been directly addressed.
>
> 1. The authors argue that their contribution lies in using a "shared continuous Neural TR illumination field" unlike the local SH in 3DGS. The reviewer find this argument unconvincing. One of the fundamental advantages of 3DGS over NeRF and the previous hybrid representations is its explicit and local nature, which allows for easy editing, compositing, scaling, and more conciseness without any neural networks. By moving the view-dependent appearance into a global, shared neural field, the method reintroduces the entanglement issues. Plenty of works before 3DGS have already taken various shared neural components, which is the most common practice at the time of NeRFs,  rather than per-primitive SH.  In my opinion, replacing SH with neural components without proving the underlying principles is actually a step backward. Using SH or a shared neural network, is just a simple choice between existing answers, but not relevant to innovations, not to say there are already plenty of 3DGS works that have used shared neural components to replace SH in the implementations. Nearly all of them have a much lower parameter amount compared to the original 3DGS.
>
> 2. The authors still fail to explain why complex scene illumination, which contains high-frequency signals like hard shadows, specularities, and occlusions, should inherently follow a low-rank, multi-linear structure. The reviewer has doubted this point in the previous review, but this concern hasn't been replied. Only showing empirical results is inadequate and weak, where plenty of uncertainties lie in.
>
> 3. Furthermore, the empirical benefit shown in the rebuttal is marginal. It remains unclear whether this specific structural bias is beyond any existing MLP-involved 3DGS (such as Scaffold-GS). Especially, Scaffold-GS is only an elder baseline for compact, high-quality 3DGS rendering. To claim superiority in "compression" and "quality", the authors must compare against similar or any better baselines. It's not an acceptable reason provided by the authors to avoid the comparison, simply speculatively stating that RingLight-GS can be combined with Scaffold-GS.
>
> 4. Regarding the comparison to EnvGS, since so far the appendix hasn't been updated, the reviewer can not fetch the details about the experiment. However, the reported scores of EnvGS from the authors are much poorer than those in EnvGS's paper. Note that the pretrained models of these two scenes are provided. These results provided in the rebuttal can not be regarded as valid, unless there is a reasonable explanation. Following the reported results by the authors, their method are even worse than the vanilla 3DGS and 2DGS in Dog of the benchmark in EnvGS.
>
> As summarized by the authors, this is an application-oriented paper, but neither provides new insights, nor comprehensive performance evaluations against any of the strong baselines. The claimed contributions on 1) performance improvement, and 2) smaller model size, are not well supported. I consider this paper substantially far from publication on ICLR, so I may maintain my recommendation.

---

> ### Author Response · Authors · 2025-11-26
> **Response to Reviewer wS37's follow-up, part 1**
>
> We thank the reviewer for the prompt reply and additional comments. We summarize your comments to the following four main concerns and will provide our point-to-point responses.
>
>  (A) Explicit/local 3DGS vs a shared neural TR illumination field,
>
>  (B) Low-rank / multi-linear structure and “lack of principles”,
>
>  (C) “Marginal” empirical benefit and “insufficient evaluation”, including Scaffold-GS, and
>
>  (D) Trust in the EnvGS comparison.
>
> **(A) On explicit/local 3DGS vs a shared neural TR illumination field**
>
> Our original motivation in the Introduction is to improve per-Gaussian color fidelity under complex illumination, enabling more compact 3DGS representations. In this submission, we focus on the appearance side of that question: starting from a standard 3DGS pipeline, we replace per-Gaussian SH by a shared Neural TR illumination field over $(x,y,z,v)$ plus an explicit base color, and evaluate whether this yields more compact models and better view-dependent quality while preserving real-time speed.
>
> In this design, the core strengths of 3DGS remain:
> - Gaussian centers, covariances, and opacities are still explicit per-primitive parameters.
> - Only the view-dependent color representation (per-Gaussian SH, which is already a non-physical, entangled parameter vector) is replaced by a structured, shared TR illumination field, with an explicit base color playing the role of albedo.
>
> Our method therefore does NOT aim to replace 3DGS as a fully editable or composable representation; it targets a specific storage–quality–runtime trade-off for novel view synthesis. In this context, we do not regard the use of a shared neural illumination field as “a step backward”, but as a design pattern that has already been used and evaluated in related works such as 3iGS and NeuralGS.
>
> Within this scope, our experiments show that the proposed shading design achieves: 1. ≈2–3× smaller models than vanilla 3DGS; 2. matched or improved PSNR/SSIM/LPIPS across three benchmarking datasets; 3. sharper and more stable specular highlights in qualitative comparisons, especially on Shiny-Blender. Note that these are achieved while maintaining real-time rendering.
>
> We view this as a meaningful improvement for the appearance representation inside a standard 3DGS pipeline. Exploring how such an improved appearance model can be combined with more aggressive Gaussian-count reduction (e.g., via pruning or anchor-based schemes) is a promising next step, but is apparently beyond the scope of this submission.
>
> **(B) On low-rank / multi-linear structure and “lack of underlying principles”**
>
> We need to first clarify an unfortunate yet obvious misunderstanding: We have never claimed that “real-world illumination is inherently low-rank and multi-linear”, nor have we tried to formally prove it in this paper. As in other tensor-based methods (e.g., TensoRF-style decompositions), we use a low-rank, factorized latent representation as a practical inductive bias and approximation, not as a strict physical law.
>
> As a result, the Neural TR illumination field never assumes that the illumination must be multi-linear in $(x,y,z,v)$. The TR contraction is multi-linear regarding the latent feature indices. In practice, each mode of the input tensor $\mathcal{B}(x,y,z,v)$ is produced by a neural TR core, so the overall mapping $(x,y,z,v)\to\mathbf{a}$ is non-linear, but with a factorized latent structure across spatial and view modes. The input $b$ is also not just a simple concatenation of ${x, y, z, v}$. This is reflected by the updated writing in Sec. 3.2.1 to avoid potential confusion.
>
> Moreover, the design pattern we adopted is motivated by literature such as [1], and we have also referred to other theoretical and empirical findings, such as [2], and introduced designs, including Fourier positional encoding, in our proposal to enable effective learning of high-frequency information.
>
> Regarding “underlying principles”, our justification follows the common pattern in NeRF/3DGS work and aligns with the well-established data-driven design and evaluation process, as reported in the manuscript. We believe this level of intuition + ablation is the standard form of architectural justification in this area.
>
> Finally, a formal theorem that complex illumination must follow a “low-rank, multi-linear structure” is way beyond the scope of this paper and is not required of related tensor- or MLP-based approaches. Therefore, we respectfully reject the request by the reviewer.
>
> [1] Luo, Y., Zhao, X., Li, Z., Ng, M. K., & Meng, D. (2023). Low-rank tensor function representation for multi-dimensional data recovery. *IEEE transactions on pattern analysis and machine intelligence*, 46(5), 3351-3369.
>
> [2] Tancik, M., Srinivasan, P., Mildenhall, B., Fridovich-Keil, S., Raghavan, N., Singhal, U., ... & Ng, R. (2020). Fourier features let networks learn high frequency functions in low dimensional domains. *Advances in neural information processing systems*, 33, 7537-7547.

---

> ### Author Response · Authors · 2025-11-26
> **Response to Reviewer wS37's follow-up, part 2**
>
> **(C) On “marginal” empirical benefit, “insufficient evaluation”, and Scaffold-GS**
>
> As in our previous responses, our paper targets a specific and, we believe, well-defined question of "Given a standard 3DGS pipeline, can we replace per-Gaussian SH by a shared, continuous TR illumination field and obtain significantly smaller models while preserving or improving the reconstruction quality with real-time speed?"
>
> All of our experiments and ablations are designed around this scope, and we respectfully disagree that the empirical benefit is marginal or insufficiently supported. Throughout our experiment, we have shown that (as in Table 1 of the manuscript): 1. Replacing per-Gaussian SH with a shared TR illumination field reduces the overall model size by ≈2–3× on Tanks&Temples, NeRF-Synthetic, and Shiny-Blender datasets; 2. On these benchmarks, our method matches or slightly improves PSNR/SSIM/LPIPS compared to 3DGS and other 3DGS-based appearance baselines such as 3iGS and GaussianShader. Qualitatively, our method produces sharper, more stable specular highlights under complex illumination, as in Figure 5. Moreover, although we introduce a shared illumination network, we still maintain clearly real-time rendering as reported in Appendix F. For a module that modifies only the color/illumination representation in 3DGS and can potentially serve as a drop-in replacement, this storage–quality–runtime trade-off is substantial and directly supported by the tables already in the paper.
>
> To avoid future confusion, **we have updated the contribution statements and the research question to improve clarity**. We believe that, with the above explanation and this update, the reviewer can now evaluate the reported results comprehensively.
>
> **Scaffold-GS** and other anchor-based approaches primarily focus on compressing geometry and attribute placement, e.g., by reducing the number of Gaussians, reorganizing primitives through scaffolds or anchors, and enabling spatial feature sharing.
> These methods alter the structural representation of the scene and are largely orthogonal to our contribution, as our method specifically targets a different component of the system: view-dependent color and illumination modeling for each Gaussian primitive, following the standard 3DGS pipeline. Our TR illumination field improves rendering quality through a more expressive radiance representation without altering the underlying geometric compression strategy. Because these two directions address different parts of the 3DGS framework, they are potentially complementary rather than mutually exclusive, and we believe this is logically acceptable.
>
> However, we still decided to conduct a direct comparison between our proposal and Scaffold-GS to directly address the reviewer’s concern. As shown in the table below and the qualitative examples in Figure 17 of the updated manuscript, our method achieves higher SSIM and PSNR, lower LPIPS, and a smaller model size on the Tank and Template scene, demonstrating that our contribution brings measurable benefits even without geometry compression.
>
> | Scene: Tank and Template | SSIM  | PSNR  | LPIPS | Size (MB) |
> |--------------------------|-------|-------|--------|-----------|
> | RingLight-GS             | 0.934 | 29.47 | 0.100  | 75.91     |
> | Scaffold-GS              | 0.927 | 28.91 | 0.108  | 83.47     |

---

> ### Author Response · Authors · 2025-11-26
> **Response to Reviewer wS37's follow-up, part 3**
>
> **(D) Trust in the EnvGS comparison**
>
> We apologize that the manuscript was not uploaded with our initial responses. Due to an unfortunate “server down” issue, we were unable to collect all experimental results (especially for the figures) before submitting our initial responses. We have now fully updated the manuscript for your reference, and we look forward to your understanding.
>
> We are also sorry that we didn’t specify the configurations in our previous response. We used the same experimental configurations as reported in our manuscript on an NVIDIA A100 GPU platform running Ubuntu 20.04. To make it more complete, we have also included comparisons with the Gaussian-Shader, 3DGS, and 3DGS-DR. The results obtained under the above settings are presented in the table below. Although pretrained weights are available for some methods, all models were retrained and evaluated from scratch on the two listed scenes using the same data split to ensure a fair comparison. The qualitative comparisons are provided in Figure 15 of the revised manuscript.
>
> **The Audi Scene**
> | Method        | SSIM  | PSNR   | LPIPS | Size (MB) |
> |---------------|-------|--------|--------|-----------|
> | RingLight-GS  | 0.861 | 27.52  | 0.197  | 197.94    |
> | EnvGS         | 0.862 | 27.55  | 0.193  | 673.34    |
> | 3DGS-DR       | 0.817 | 25.91  | 0.205  | 276.95    |
> | G-shader      | 0.804 | 25.36  | 0.213  | 378.41    |
> | 3DGS          | 0.837 | 27.01  | 0.199  | 444.54    |
>
> **The Dog Scene**
> | Method        | SSIM  | PSNR   | LPIPS | Size (MB) |
> |---------------|-------|--------|--------|-----------|
> | RingLight-GS  | 0.871 | 24.82  | 0.213  | 279.43    |
> | EnvGS         | 0.869 | 24.80  | 0.213  | 1264.25   |
> | 3DGS-DR       | 0.848 | 22.87  | 0.247  | 493.39    |
> | G-shader      | 0.854 | 23.58  | 0.235  | 668.30    |
> | 3DGS          | 0.868 | 24.69  | 0.217  | 597.58    |
>
> Note that it is very common for cross-paper re-evaluations to deviate from the original published numbers due to differences in metric implementations, data cropping or alignment, and library versions and hardware. The same applies if other configuration details differ. With our configurations specified, we believe the comparisons are valid.
>
> At the same time, **our central claims do not rely on outperforming EnvGS on its own benchmark**. To recap, EnvGS introduces an additional environment Gaussian field and a ray-tracing renderer built on OptiX and RT cores to explicitly model environment reflections. Our work remains in the standard 3DGS rasterization framework and focuses on a different trade-off, as in our above response. These performance results are thus intended to provide some potential insights in response to your request.
>
> In addition, we intended to use our benchmarking datasets for a more formal comparison with EnvGS. However, after applying the provided data pre-processing steps in the EnvGS repository, we were unable to achieve meaningful training for EnvGS, leading to the fallback alternative we introduced above for proof-of-concept comparisons.
> This also supports the idea that EnvGS and our proposal are implemented in different pipelines (unlike Scaffold-GS, whose implementation requires minimal modification to run with our configurations), which is a non-trivial factor to consider in real-world applications.
>
>
>
> -----------
> To summarize, we reiterate that our paper is application-oriented rather than a theoretical breakthrough. However, within that scope, we believe the contribution is non-trivial and well supported, and we have provided additional explanations with evidential support to address the reviewer’s concerns directly.
>
> Therefore, we respectfully maintain that our work constitutes a solid, practically relevant contribution to the 3DGS and NVS literature, given the clearly stated scope and the experimental evidence.

---

> ### Comment · Reviewer_wS37 · 2025-11-26
>
> Thanks for the details about the experiments. However, the major defects of this work, about *without proving the underlying principles*, still remain and have not been addressed. In my opinion, being responsible for the correctness of the claim contribution is a basic requirement of an academic paper. When *the rationale of the proposed module is in conflict with the solved problem claimed by the authors*, it's a **strongly irresponsible behavior** to reject offering any reliable proof and explanation in principle. It's a basic responsibility for the authors and reviewers to ensure the correctness of the information delivered to the community, while the authors failed to do so. What the authors have repeatedly done is just describe the technical design, which can not derive how the improvements come. This is my main concern about this work.
>
> In the part of experiments, given that there are too many tricks or untrusted ways to benefit the metrics on various benchmarks, it's extremely weak to just provide empirical results as the only support. It can make the contributions overclaimed. I sincerely appreciate the new experiments conducted by the authors, but after carefully reviewing the paper once again, I find that the setting of Tanks and Temples is confusing, of which the reported scores can not match any of the prevailing settings used in other papers that are ensured convincing and reproducible. Meanwhile, many of the baselines' metrics on NeRF-Synthetic and Shiny-Blender are also much lower than reported in other papers, including some of which are actually reproducible as far as I know. It's not an acceptable explanation of the performance gap up to 1-2 PSNR just because of environmental differences. Therefore, I do not consider the experiments, including those complemented in the rebuttal, are strong and convincing enough.
>
> Besides, there are still other problems like the less novel idea and lack of a comprehensive comparison, where the Scaffold-GS and EnvGS are the lower bound as I have described, but not the only lacking methods. But I think they are less important than the two major problems mentioned above. It's suggested to conducing experiments on standard benchmarks with widely accepted baseline metrics to reduce the uncertainties introduced during the less transparent reproduction. Overall, I consider this paper is still far from the bar of publication on ICLR this time, for its critical defect of soundness.

---

> ### Author Response · Authors · 2025-11-28
> **Final clarification on scope, soundness, and experiments**
>
> We thank the reviewer for the speedy reply and additional comments. At this point, we would like to briefly clarify our position rather than repeat arguments and results already presented in detail above.
>
> **1. On "underlying principles" and soundness.**
>
> After a minor writing update, our paper now explicitly makes an empirical and bounded claim: within a standard 3DGS pipeline and a fully specified training/evaluation setup, replacing per-Gaussian SH with our TR-based illumination representation yields a more favorable storage–quality–speed trade-off (smaller models, equal or better metrics, real-time FPS).
>
> We have not, and still do not, claim that illumination is inherently low-rank or multi-linear, nor that our module is theoretically necessary or optimal; TR is presented as a data-driven architectural choice, justified in the usual way in this literature—by hypothesis, targeted ablations, and grounded comparison against reasonable alternatives. With this clarification, we believe the rationale of the proposed module is consistent with the problem we claim to address, contrary to the reviewer's interpretation.
>
> The reviewer's insistence on a "formal proof of correctness" of the architecture itself **goes substantially beyond the norm in NeRF/3DGS and other deep learning work at ICLR (and related prestigious conferences/journals)**. We respectfully disagree with the conclusion that the absence of such proof, for an empirically validated module, renders the paper "unsound".
>
>
> **2. On empirical evidence and additional experiments.**
>
>  Following the reviewer's requests, we have run additional baselines (including EnvGS and Scaffold-GS) under a single, transparent evaluation pipeline, clarified our settings, and shown consistent relative gains under that shared setup. The reviewer did not identify any concrete errors in these experiments, but instead expressed general distrust of the empirical results simply because our absolute numbers do not match previously published metrics for these datasets. This is, unfortunately, a very weak and vague rationale. In practice, 1–2 dB differences across papers are common due to differences in resolution, cropping, tone mapping, metric variants, and other factors. That is precisely why we base our claims on relative performance within one reproducible pipeline, which is standard practice in this area. The same reasoning applies to our attempts to evaluate EnvGS on our own datasets, as reported above, despite an unsuccessful attempt following the provided steps.
>
>  We stand by the correctness of the reported experiments and believe they support the **now clearly scoped claims**, and **we do not find it appropriate to characterize our experimental protocol as "tricks"**, particularly in the absence of any identified technical errors.
>
>
> **3. On tone and framing.**
>
> We appreciate rigorous criticism and have revised the manuscript accordingly (research question, contribution statement, clarification of assumptions, and new baselines, **all in line with the reviewer's earlier requests**). We believe our responses have addressed the technical concerns as far as possible within this compact rebuttal phase.
>
> Therefore, we disagree with the characterization that presenting a carefully evaluated, empirically motivated architecture— simply without a formal optimality proof of the architecture itself—constitutes a "strongly irresponsible behavior". **Such phrases frame the discussion in moral rather than technical terms**, which we consider **highly inappropriate** for evaluating the work's technical content.
>
>
> --------
> We will leave it to the ACs to judge whether the level of technical contribution and experimental evaluation is in line with prevailing ICLR standards in this area.

---

> ### Comment · Reviewer_wS37 · 2025-11-28
>
> **I'm sad to see the authors try to escape their responsibility for an academic paper by overclaiming my request**. In fact, I have never stated a "formal proof of correctness" is required. The word of "formal proof" was distortedly added by the authors in their second round rebuttal. Here I show my original words:
>
> > The authors still fail to explain why complex scene illumination, which contains high-frequency signals like hard shadows, specularities, and occlusions, should inherently follow a low-rank, multi-linear structure. The reviewer has doubted this point in the previous review, but this concern hasn't been replied. Only showing empirical results is inadequate and weak, where plenty of uncertainties lie in.
>
> It's clear that a reasonable explanation is what I consider necessary for the soundness of this paper, but not the distorted, stricter "formal proof" I have never mentioned. When **the actual illumination could be non-linear and complex to describe but the authors counterintuitively stated that a low-rank, multi-linear structure can better capture that than existing non-linear networks**, it's an essential part for the soundness to provide why it can happen in rationale. **This should not naturally work in principle, and so only empirical results are not sufficient in this special case**.
>
> In the part of the experiments, after reviewing my last round reply, I first express my apologies for the misleading words in "In the part of experiments, given that there are too many tricks or untrusted ways to benefit the metrics on various benchmarks, it's extremely weak to just provide empirical results as the only support. ". Here I mean there are many potential tricks these days, but not indicate the authors have used them. What I want to deliver is the part of "it's extremely weak to just provide empirical results as the only support.". Still, I do not consider the experiments convincing enough, and **do not think the huge gap of  1-2 PSNR is a reasonable deviation when the results can be reproduced by third parties**. And also as I mentioned, **the used settings, such as selected scenes and reproduction details of each baseline, are not given**, especially when the metrics are significantly biased from the other sources, which raises a transparency problem and fairness concerns.
>
> Note that these are the critical concerns of the paper quality itself, but not the personal requests from the reviewer. After several rounds of discussions, the authors still refused to directly address the mentioned two major concerns.
>
> According to the ICLR 2026 reviewer guideline, the reviewers are requested to consider the critical question during the review:
> > Answer four key questions for yourself, to make a recommendation to Accept or Reject:
> >
> > 3. Does the paper support the claims? This includes determining if results, whether theoretical or empirical, are correct and if they are scientifically rigorous.
>
> This is the basic requirement of the paper for ICLR. The provided evidence in this paper can not support its claims. The methodology is less sound, conditions are not well controlled with large biases and uncertainties, and the transparency is poor, therefore the paper is not scientifically rigorous. I'll keep my recommendation to reject, and will respect the recommendation from the AC and other reviewers.

---

> > ### Author Response · Authors · 2025-11-28
> > **Final remarks to Reviewer wS37**
> >
> > Our final clarification to the reviewer.
> >
> > 1. **We have not, and still do not, claim that “complex illumination inherently follows a low-rank, multi-linear structure”.** Our claims are bounded and supported by empirical findings; requests to justify stronger theoretical statements go beyond the scope of what we have stated in the paper.
> >
> > 2. **Comprehensive empirical evaluation with a clearly specified training and evaluation setup is standard practice in NeRF/3DGS and, more broadly, deep learning research.** We have reported our experimental setups truthfully in the manuscript and in our responses to the reviewer’s previous comments, and our conclusions are supported by the relative performance within this unified, reproducible pipeline, which **we will make open-source for public evaluation**.
> >
> > We have no further technical clarifications to add and respectfully leave the final judgement to the ACs.

---

> > > ### Comment · Reviewer_wS37 · 2025-11-28
> > >
> > > It seems the authors still failed to get what the major problems are. To help the authors understand the reviews to improve their paper, in summary, here are two points:
> > >
> > > 1) The authors failed to explain why their work can work.
> > >
> > > 2) The details of experiment settings are not given, and the experimental results are significantly biased from third-party results.
> > >
> > > I don't know why the authors always try to avoid directly giving the answers, but repeatedly distort my partial words. From the reviewer and a reader's perspective, the correctness can not be ensured when the two things happen simultaneously: 1) the methodology is not proven (just a note: this does not indicate a "formal proof") to be sound; 2) experiments are neither transparent, nor close to those reproduced by third-party works. It's not the case of whether the experiment problems were unintentionally made or not. Providing code in the future is not a solution for it.
> > >
> > > If there is any existing answer, please directly place it exactly as is to help the AC's decision, rather than just stated that these have been provided. Giving a sound explanation about why it can work is a basic requirement and **can not be out of scope for any paper**.

---

### Official Review · Reviewer_YrpD · 2025-10-29

**Soundness:** 3
**Presentation:** 3
**Contribution:** 3
**Rating:** 6
**Confidence:** 4

**Summary:**

The paper proposes RingLight-GS, a compact extension of 3D Gaussian Splatting that separates scene color into a view-independent base and a view-dependent residual. By modeling residuals with a neural tensor ring regression conditioned on position and view direction, the method improves rendering quality under complex illumination while significantly reducing storage requirements.

**Strengths:**

The paper presents a novel approach to modeling view-dependent appearance in Gaussian Splatting using Neural TR Regression, which dynamically generates high-dimensional tensor slices for each spatial position and view. This method is original in combining tensor decomposition with neural feature learning, effectively capturing complex lighting effects without explicit BRDF parameters. The approach is well-motivated, clearly explained, and demonstrates potential for high-quality, compact scene representation, addressing key limitations of prior SH-based methods.

**Weaknesses:**

- While the Neural TR Regression effectively models view-dependent appearance with compact storage, it introduces additional model complexity and relies on high-dimensional tensor decomposition, which may increase training time and memory usage. Furthermore, the approach assumes accurate input positions and viewing directions, making it potentially sensitive to noise or imperfect geometry.

- The presentation of experimental results could benefit from supplementary video demonstrations (e.g., depth maps, normal maps). I strongly encourage the authors to include video comparisons on test scenes to better showcase the effectiveness of the proposed method.

- Can this framework be extended to 2D Gaussian Splatting (2DGS)? I am curious about the performance implications if 2DGS were used instead of 3DGS. If I understand correctly, subsequent designs such as the “Neural TR decomposition” appear to be agnostic to the choice of primitive representation. How would the rendering quality compare if 2DGS is adopted? If 2DGS cannot be directly integrated into the current framework, what are the main limitations preventing this?

**Questions:**

See Weaknesses

---

> ### Author Response · Authors · 2025-11-21
> **Response to Reviewer YrpD**
>
> We appreciate Reviewer YrpD's recognition of our work and insightful comments for further improvement. We now respond to the reviewer's comments in a point-by-point manner.
>
> **[W1] Model Complexity and Resource Cost of Neural TR Regression**
>
> We acknowledge that introducing the Neural TR module complicates the overall model design. However, the purpose of TR decomposition is to replace the expensive per-Gaussian SH coefficients in 3DGS with a shared, compact, and expressive representation, resulting in substantial reductions in memory usage and improved rendering quality. As shown in our experiments (e.g., Table 1 and Appendix F), inference remains real-time, while the model size is significantly smaller than prior methods with similar fidelity, despite moderately prolonged training.
>
> **[W2] Robustness to Noise in Geometry and Camera Input**
>
> We appreciate the reviewer’s point regarding sensitivity to input accuracy. Like all 3DGS-based methods, RingLight-GS operates on fixed input point positions and camera parameters, and thus shares the same baseline assumption of accurate geometry and camera calibration for best practice. However, the datasets used in our experiments (Tanks and Temples) inherently contain imperfect reconstructions, occlusions, and calibration errors.
>
> The stable performance we observe in these settings indicates that our method remains robust under typical real-world noise levels. As a result, our method does not appear to be sensitive to natural noise in the input. The use of a shared illumination field (via neural TR) and the explicit disentanglement of base and residual color introduce a degree of structural regularization that, in theory, helps reduce overfitting to noisy signals.
>
> In real life, noise reduction and camera calibration are complex topics that can be resolved through both hardware and software approaches. Within the software domain, noise reduction can be achieved through efficient filtering and data engineering during the preprocessing stage (which can be added as an independent module to our framework). However, this aspect is not the focus of our work, and we believe our elaboration can address the reviewer’s concern.
>
> **[W3] Supplementary Visualizations and Video Demonstrations**
>
> We thank the reviewer for the constructive suggestion and agree that supplementary video demonstrations can further enhance the presentation and help highlight the strengths of our method. We plan to include such video comparisons as part of our public code release. The reason is that such a comparison does not affect the main findings of our work, whereas a well-structured video comparison requires engineering work that may not be feasible within the limited time of the rebuttal. We look forward to the reviewer’s understanding.
>
> **[W4] Applicability of Neural TR and Residual Color Modeling to 2D Gaussian Splatting**
>
> We thank the reviewer for this thoughtful and forward-looking question. Our method is built on top of 3DGS because all main baselines we compare to (3DGS, 3iGS, GaussianShader, Glossy-GS, LightGaussian, etc.) are 3DGS-based and use per-Gaussian SH, which is exactly what we aim to replace. The proposed Neural TR illumination field itself is primitive-agnostic: it only takes continuous 3D position and view direction $(x,y,z,v)$ and outputs appearance features, so in principle it could also be plugged into a 2DGS-style renderer as an alternative to SH/MLP-based color.
>
> However, we need to note that our method for 2DGS is NOT a direct drop-in change: 2DGS uses a different primitive parameterization and rendering backend, and its memory layout and visibility handling differ from 3DGS. Our storage and speed analysis is tied to the standard 3DGS attribute layout, where SH dominates memory; repeating all experiments on a 2DGS implementation would require a substantially different codebase and a new set of baselines. For these reasons, we consider the application of our method to 2DGS a promising but orthogonal piece of work that is beyond the scope of this paper. We believe this explanation clears out the reviewer's confusion.

---

### Official Review · Reviewer_UJPX · 2025-11-01

**Soundness:** 2
**Presentation:** 2
**Contribution:** 2
**Rating:** 4
**Confidence:** 2

**Summary:**

This paper presents RingLight-GS, a method that enhances 3D Gaussian Splatting (3DGS) by improving compactness and handling of complex lighting effects. Traditional 3DGS relies on large Spherical Harmonics (SH) coefficients to represent color, leading to high memory consumption. RingLight-GS addresses this by decomposing each Gaussian’s color into a view-independent base color and a view-dependent residual color. The core contribution is a Neural Tensor Ring (TR) Regression model that learns the view-dependent component. This TR model maps spatial coordinates and viewing directions to appearance features through a decomposed tensor representation that is both compact and expressive. As a result, RingLight-GS achieves approximately 2.5–3× storage reduction compared to standard 3DGS, while improving rendering quality, particularly for specular highlights and reflections.

**Strengths:**

1. Separating the base color from a tensor-factorized residual is an elegant and effective design. It reduces 3DGS’s memory overhead and improves its ability to represent complex, view-dependent effects like specular highlights that SHs handle poorly.

2. The method achieves both compactness and improved performance: it reduces model size substantially (e.g., from 68 MB to 22 MB on NeRF-Synthetic) while also attaining higher PSNR, SSIM, and LPIPS scores across multiple datasets, outperforming existing compression and lighting-aware 3DGS variants.

**Weaknesses:**

The advanced Neural TR model incurs a computational cost. The paper reports substantially longer training times (e.g., ~29 minutes versus ~15 minutes for standard 3DGS on Shiny-Blender) and increased VRAM usage during training, making it less efficient to train and deploy than the original 3DGS.

The method introduces several new components (neural TR, render module, loss scaling factor γ) and associated hyperparameters (TR rank, feature dimension). The ablation studies show performance degrades if these are not set correctly, adding complexity and potentially limiting its ease of adoption.

In Figure 2, which aims to illustrate the overall scene color modeling framework, is too schematic and abstract. It uses generic blocks (e.g., "TR Function Regression," "Render Module") without visually hinting at their unique internal mechanics (e.g., how the neural TR cores connect). This makes it difficult for a reader to grasp the innovative data flow and architecture without constantly cross-referencing the complex text in Section 3.2. A more detailed or annotated diagram would be greatly beneficial.

**Questions:**

The Neural TR module plays a central role in modeling view-dependent effects. It remains unclear how its efficiency and compactness scale with scene complexity. In scenes with substantially more Gaussians or a larger spatial extent, would preserving rendering quality necessitate a proportional increase in TR rank or feature dimension, thereby diminishing the storage benefits?

---

> ### Author Response · Authors · 2025-11-21
> **Response to Reviewer UJPX**
>
> We sincerely thank Reviewer UJPX for the constructive feedback on our method. We now provide our point-to-point response to the weaknesses and questions posted by the reviewer below.
>
> **[W1] The advanced Neural TR model incurs a computational cost**
>
> We acknowledge that the neural TR module introduces additional overhead during training, resulting in longer training time compared to standard 3DGS. As already noted in the Conclusion of the submitted paper, reducing the neural TR overhead is an important direction for our future work.
>
> However, we would like to clarify that this has a minimal impact on the inference stage for deployment. RingLight-GS maintains the real‑time rendering performance of 3DGS and is significantly faster than NeRF-based approaches. Our proposal also substantially reduces storage requirements while producing consistently higher visual quality. The benefits in compactness and rendering fidelity far outweigh the additional overheads, mainly applicable to the training stage, making the trade-off highly favorable. We invite the reviewer to refer to Appendix F for more numerical details.
>
> **[W2] Concern about the introduction of new components and associated hyperparameters**
>
> We have conducted a comprehensive analysis of the new components’ influence in Appendix E. As shown in the tables there, our default settings (TR rank = 16, appearance dimension = 48) consistently yield strong performance across all datasets, striking a balance between quality, training time, and rendering speed. The results also show that only an extremely low TR rank or feature dimension significantly affects the expressiveness of the neural TR module; performance tends to stabilize once the parameters are within a reasonable range, indicating robustness to moderate variation.
>
> **[W3] Figure 2 is too schematic and abstract**
>
> We understand your concern that the original Figure 2 may not sufficiently convey the internal data flow or structural details of the proposed scene color modeling framework. We have revised this figure to improve the flow of data/feature flow. Also note that Figure 4. b further illustrates the composition of neural TR for 3-D inputs, which may further help your understanding of Figure 2's corresponding part.
>
> **[Q1] Scalability and Storage Efficiency of Neural TR under Complex Scenes**
>
> In our design, the primary source of memory usage comes from per-Gaussian attributes such as opacity, while the neural TR parameters and MLP are shared across all Gaussians and do NOT scale with scene size. Consequently, even in scenes with significantly more Gaussians or spatial complexity, increasing the TR rank or feature dimension does NOT proportionally increase total memory usage. As our response to W2, model performance tends to stabilize once these parameters reach a moderate level, and further increases yield diminishing returns (shown in Appendix E). More importantly, our approach replaces the large per-Gaussian SH vectors with a compact, expressive neural field, resulting in substantial memory savings regardless of scene size. Thus, RingLight-GS preserves its scalability and storage efficiency even in complex environments.
>
> To further verify scalability under complex conditions, we evaluated our method on three complex real-captured scenes (bicycle, bonsai, counter) from the Mip-NeRF 360 dataset [1], which exhibit greater scene extent, natural lighting, and specular effects. The table below shows that RingLight-GS achieves the best rendering quality across all metrics. We also include rendered images for qualitative comparison in the newly added Figure 14 in Appendix G for your further reference.
>
> | Method              | SSIM  | PSNR  | LPIPS | Size (MB) |
> |---------------------|-------|-------|-------|-----------|
> | 3iGS                | 0.895 | 29.62 | 0.179 | 374.52    |
> | G.shader            | 0.897 | 29.32 | 0.176 | 311.70    |
> | Glossy-GS           | 0.893 | 29.81 | 0.181 | 412.53    |
> | 3DGS-DR             | 0.886 | 29.57 | 0.184 | 286.74    |
> | **RingLight-GS (Ours)** | 0.904 | 30.17 | 0.171 | 101.98    |
>
>
> [1] Barron, J. T., Mildenhall, B., Verbin, D., Srinivasan, P. P., & Hedman, P., “Mip-NeRF 360: Unbounded Anti-Aliased Neural Radiance Fields,” in Proceedings of the IEEE/CVF Conference on Computer Vision and Pattern Recognition (CVPR), 2022, pp. 5470-5479.

---

### Author Response · Authors · 2025-11-30
**Authors' Final Remarks on the Rebuttal**

## Reviewers UJPX, YrpD, DfdE

None started further discussions since the rebuttal; None questioned our method's soundness. All were mildly positive.

### UJPX (4, "low confidence")

**Initial stance**
Found our idea of design elegant and the compression/quality trade-off interesting, but was concerned that:

- TR module adds training cost \& VRAM usage.
- Extra components/hyperparameters may hurt simplicity \& robustness.
- Scalability of the approach to larger/more complex scenes was unclear.

**Our response**
We clarified that:

- The main overhead is in training, while inference remains real-time and significantly faster than NeRF-based NVS, with 2-3× model-size reduction at comparable or better quality.
- TR parameters are shared \& do not scale with the number of Gaussians.
- Performance is stable across a moderate range of TR rank \& feature dimension (Appendix E).
- We added results on more challenging scenes and improved schematic figures to better illustrate the internal flow.

### YrpD \& DfdE (both 6, "medium confidence")

**Initial stance**
Both considered the work technically sound and clearly written.  Their main concerns are:

- Training-time overhead \& complexity.
- Hyperparameter choices.
- How stable the base/residual decomposition is on diffuse \& specular scenes.
- A conceptual question about possible extension to 2DGS.

**Our response**

- Making explicit that the contribution is an improvement in the storage–quality trade-off within a standard 3DGS pipeline, not a new 3D representation.
- Detailing γ, TR rank, feature dimension, training schedule \& optimizers (Sec. 4.1, Appendix A/E), and noting that β is inherited from 3DGS as a constant for fairness.
- Adding multi-view visualizations that show consistent base/residual behavior across views \& across diffuse/specular scenes;
- Clarifying that 2DGS extension is theoretically viable and a potential future work.

---

## Reviewer wS37 (2, "high confidence")

**Initial stance**
wS37 raised broad concerns about “soundness”:

- Base + residual color separation is not original \& essentially equivalent to SH in 3DGS.
- Neural TR is “not different” from a NeRF-style MLP and has no meaningful inductive bias.
- Ablations are “inconclusive” with small improvements.
- Baselines like EnvGS, Scaffold-GS, and newer 3DGS compression works are missing.

**Our response**
We revised and clarified the paper:

- **Scope and claims.** Explicitly frame our contribution as an empirical, bounded improvement within a standard 3DGS pipeline with previous theoretical support. We updated the research question and contribution statement accordingly.
- **Technical clarification**
  - Explain that the shared TR field over $(x,y,z,v)$ differs from per-Gaussian SH \& from concatenated-input MLPs, and state clearly that we do not claim that illumination inherently follows a low-rank, multi-linear structure, nor does our design rely on it.
  - Clarified that Sec. 4.3 \& Appendix B already provides comparisons between TR, NeRF-style MLP, and other tensor decompositions.
- **Experimental protocol \& baselines**
  - Added *EnvGS* and *Scaffold-GS* baselines (as requested) with unified configurations where possible.
  - Clarified that RingLight-GS consistently achieves good performance to model-size trade-off and remains real-time.

**Later comments \& remaining disagreement**
In the follow-ups, wS37:

- Continues to argue that our work is “not (theoretically) proven to be sound”, although our clarified empirical and bounded claims and the provided ablations follow the norms in this field.
- States that “detailed experiment settings not given” even though Sec. 4.1 \& Appendix A/E/F explicitly report on datasets, splits, metrics \& hyperparameters.
- Expresses general distrust simply because some absolute PSNR/SSIM values do not fully match other papers, but does not identify specific experimental errors.
- Uses value-laden language, e.g., “strongly irresponsible behavior”.

We post a short final clarification, reiterating that:

 - We never made the “inherently low-rank, multi-linear illumination” claim.
- Our claims are empirical \& bounded.
- The evaluation protocol is fully specified and source code will be released.
- We have no further technical clarifications to add.

---

Three reviewers find the paper technically sound and practically useful; Concerns focused on details such as complexity, hyperparameters, costs, and presentation, which we have addressed through clarifications and additional experiments.

The remaining disagreement with wS37 is not about a concrete error or an undisclosed limitation, but about an expectation for theoretical proofs and absolute metric matching much stronger than the typical practice, despite our clearly specified \& unified protocol, bounded claims, supported by previous research.

We respectfully leave the final decision to the ACs and trust your fair evaluations.

---

### Note · Authors · 2026-03-03

I have read and agree with the venue's withdrawal policy on behalf of myself and my co-authors.

---

### Meta-Review · Area_Chair_Jgkg · 2025-12-29

**Summary:**

This paper presents RingLight-GS, a compact framework for modeling scene color in 3D Gaussian Splatting (3DGS) that addresses storage demands and view-dependent appearance under complex illumination. The work separates scene color into a view-independent base color and a view-dependent residual color, with the residual modeled via Neural Tensor Ring (TR) decomposition conditioned on spatial positions and viewing directions. The three main contributions are: (1) explicit base/residual color decomposition tailored to 3DGS; (2) Neural TR regression for compact, shared illumination representation over (x,y,z,v); and (3) approximately 2-3× storage reduction while maintaining or improving rendering quality.

The paper received scores of 4, 6, 6, and 2 from four reviewers (`UJPX`, `YrpD`, `DfdE`, `wS37`). RingLight-GS achieves 2.5-3× storage reduction compared to standard 3DGS while improving or matching PSNR/SSIM/LPIPS across Tanks&Temples, NeRF-Synthetic, and Shiny-Blender datasets. Qualitative results demonstrate sharper specular highlights and better material consistency. Authors provided comprehensive responses including additional baselines (EnvGS, Scaffold-GS), ablations on tensor decompositions (CP, VM), hyperparameter analysis (Appendix E), multi-view consistency visualizations (Appendix G Figure 16), and experiments on complex real-world scenes (Mip-NeRF 360).

**Reviewer Concerns:**

**Addressed concerns**:

Reviewer `UJPX` raised concerns about computational cost, hyperparameter complexity, scalability, and schematic figures. Authors clarified that training overhead exists but inference remains real-time with significantly faster performance than NeRF-based methods while achieving 2-3× model-size reduction; TR parameters are shared across all Gaussians and do not scale with scene size; performance is stable across moderate ranges of TR rank and feature dimension (Appendix E); and they improved Figure 2 and added results on complex Mip-NeRF 360 scenes (bicycle, bonsai, counter) showing best rendering quality across all metrics with smallest model size (101.98 MB vs 286-412 MB for baselines).

Reviewer `YrpD` questioned model complexity, resource costs, robustness to input noise, need for video demonstrations, and applicability to 2DGS. Authors explained that TR decomposition replaces expensive per-Gaussian SH with shared compact representation yielding substantial memory reduction and improved quality despite moderately prolonged training; the method shares baseline assumptions of accurate geometry/camera calibration with all 3DGS-based methods but demonstrates robust performance on datasets with inherent imperfect reconstructions (Tanks&Temples); they plan to include video comparisons in public code release; and clarified that 2DGS extension is theoretically viable but requires substantially different codebase and new baselines, making it promising future work beyond current scope.

Reviewer `DfdE` requested hyperparameter analysis and multi-view consistency evidence. Authors provided detailed explanations of γ (scaling factor set to 10,000, stable for 5,000 < γ < 15,000) and β (inherited from 3DGS, not a new hyperparameter), with comprehensive analysis in Appendix E; added multi-view renderings (Appendix G Figure 16) demonstrating spatial and angular coherence; showed stable base/residual decomposition across both glossy (drums) and diffuse (train) materials in Appendix D Figure 9; confirmed effectiveness on predominantly diffuse Tanks&Temples outdoor scenes; and explained that training overhead comes from dynamic tensor-ring core generation but inference remains efficient and real-time.

**Outstanding concerns**:

Reviewer `wS37` maintained a score of 2 with high confidence, raising concerns about theoretical justification and experimental validity. The reviewer requested explanation of why complex scene illumination with high-frequency signals (hard shadows, specularities, occlusions) should follow a low-rank, multi-linear structure. Authors clarified that they make no claim about illumination being inherently low-rank/multi-linear; rather, TR is used as a practical inductive bias and approximation, similar to other tensor-based methods like TensoRF. The disagreement centers on whether empirical validation with ablations suffices, or whether stronger theoretical justification is required.

The reviewer expressed concerns about experimental transparency, noting that reported metrics deviate 1-2 dB from other papers and that detailed experiment settings for scene selection and baseline reproduction are not fully specified. Authors responded that cross-paper metric variations are common due to differences in resolution, cropping, tone mapping, and library versions, and that their claims are based on relative performance within a unified pipeline. They provided additional comparisons (EnvGS, Scaffold-GS, CP/VM decompositions) with specified configurations.

The core methodological question remains: whether the base+residual separation via shared TR field represents a meaningful architectural contribution beyond existing approaches (per-Gaussian SH in 3DGS, shared neural components in prior works). The reviewer views it as combining existing techniques without clear principles; authors frame it as an effective empirical solution achieving favorable storage-quality-speed trade-offs. The performance improvements shown (2-3× compression with matched/improved metrics) are factual, but whether this constitutes sufficient contribution for ICLR acceptance remains a judgment call on incremental versus substantial advance.

**Reviewer Scores:**

**Current Scores:**
- **Reviewer `UJPX`**: 4 (marginally below threshold, low confidence) - found design elegant and compression/quality trade-off interesting; concerns about training cost, complexity, and scalability addressed with clarifications and additional experiments
- **Reviewer `YrpD`**: 6 (marginally above threshold, medium confidence) - considered work technically sound and clearly written; concerns about complexity, robustness, and 2DGS extension addressed with detailed explanations
- **Reviewer `DfdE`**: 6 (marginally above threshold, medium confidence) - found approach original with solid technical work and comprehensive evaluation; concerns about hyperparameters and multi-view consistency addressed with analysis and visualizations
- **Reviewer `wS37`**: 2 (reject, high confidence) - maintained fundamental objections about soundness and experimental validity despite comprehensive author responses

**Expected post-discussion scores**: 4, 6, 6, 2 (median: 5)

---

### Decision · Program_Chairs · 2026-01-26

Reject